# Larger and more instructable language models become less reliable

Lexin Zhou[1,2], Wout Schellaert[1,3], Fernando Martínez-Plumed[1,4], Yael Moros-Daval[1], Cèsar Ferri[1,4] & José Hernández-Orallo[1,3,4 ✉]

The prevailing methods to make large language models more powerful and amenable have been based on continuous scaling up (that is, increasing their size, data volume and computational resources[1]) and bespoke shaping up (including post-filtering[2,3], fine tuning or use of human feedback[4,5]). However, larger and more instructable large language models may have become less reliable. By studying the relationship between difficulty concordance, task avoidance and prompting stability of several language model families, here we show that easy instances for human participants are also easy for the models, but scaled-up, shaped-up models do not secure areas of low difficulty in which either the model does not err or human supervision can spot the errors. We also find that early models often avoid user questions but scaled-up, shaped-up models tend to give an apparently sensible yet wrong answer much more often, including errors on difficult questions that human supervisors frequently overlook. Moreover, we observe that stability to different natural phrasings of the same question is improved by scaling-up and shaping-up interventions, but pockets of variability persist across difficulty levels. These findings highlight the need for a fundamental shift in the design and development of general-purpose artificial intelligence, particularly in high-stakes areas for which a predictable distribution of errors is paramount.

Millions of people are using general-purpose artificial intelligence (AI) systems based on large language models (LLMs), which have become commonplace in areas such as education[6], medicine[7], science[8,9] and administration[10,11]. As these models frequently make mistakes, users have to supervise model operation and manage their expectations, for the reliable use of these systems. With language models becoming larger and more instructable, we need to analyse how this reliability has evolved. Since the early LLMs[12–14], models have been scaled up—trained with more parameters, on larger datasets and with longer training times—and have also been shaped up with human feedback—using techniques such as instruction fine tuning[4], reinforcement learning from human feedback (RLHF)[5] or output-filtering moderation techniques[2,3].

It may be taken for granted that as models become more powerful and better aligned by using these strategies, they also become more reliable from a human perspective, that is, their errors follow a predictable pattern that humans can understand and adjust their queries to[15]. For instance, early models failed at simple additions such as '20 + 183'. Performance was highly predictable: failure was common. As a result, users easily understood that there was no operating range for this task: nobody used these models for addition. A few scaled-up and shaped-up generations later, the models not only seemingly master these additions but also successfully perform additions of 50 digits or more. Because of this prowess, people may start using them as calculators (for example, to convert measurements to different units[16]). It is only in such cases that users become disappointed when the model fails at a

simple prompt such as 'Add 3913 and 92'. The user-driven reliability is then seriously damaged, because the model fails when the user thinks these digits were in the operating range. The experience becomes even more baffling when the user gets the correct answer if the question is adjusted slightly, for example to '3913 + 92 =', or if it is not changed at all—because many models are configured to be non-deterministic. Although this prompt sensitivity has been analysed extensively[17–20], it is poorly understood why an over-diligent system spouts a wrong answer for 100-digit addition instead of simply answering 'I'm afraid I can't do that'. This reckless behaviour has been incentivized by developers building models that are 'never evasive'[21].

## Reliability fluctuations

To understand the evolution of reliability, we analyse the trajectory of several families of LLMs: the generative pre-training (GPT) saga developed by OpenAI, the LLaMA series developed by Meta and the BLOOM suite developed by BigScience. GPT has led the state of the art in the past few years and, according to several surveys[22–24], is central to the LLM ecosystem, influencing transformer-based architectures, training data, evaluation frameworks and alignment techniques. LLaMA[25,26] is the best example of a family for which weights have been released, and BLOOM[27,28] is the result of an even more open endeavour coming from the scientific community. Each family represents a genuine effort of making LLMs more capable and better aligned at

[1]Valencian Research Institute for Artificial Intelligence (VRAIN), Universitat Politècnica de València, Valencia, Spain. [2]University of Cambridge, Cambridge, UK. [3]Leverhulme Centre for the Future of Intelligence, University of Cambridge, Cambridge, UK. [4]ValGRAI, Valencia, Spain. ✉e-mail: jorallo@upv.es

**Table 1 | Ten GPT, ten LLaMA and twelve BLOOM models**

| Model | Release year | Scaling | | | Shaping | |
|---|---|---|---|---|---|---|
| | | Size (no. of parameters) | Data (no. of tokens) | Compute (no. of FLOPs) | Instruction | Alignment |
| GPT-3 ada | 2020 | 350 M | 300 B | $6.41 \times 10^{20}$ | None | None |
| GPT-3 babbage | 2020 | 1.3 B | 300 B | $2.38 \times 10^{21}$ | None | None |
| GPT-3 curie | 2020 | 6.7 B | 300 B | $1.20 \times 10^{22}$ | None | None |
| GPT-3 davinci | 2020 | 175 B | 300 B | $3.14 \times 10^{23}$ | None | None |
| text-davinci-001 | 2021 | 175 B | – | – | FeedME | None |
| text-davinci-002 | 2022 | 175 B | – | – | FeedME | None |
| text-davinci-003 | 2022 | 175 B | – | – | RLHF (PPO) | None |
| GPT-3.5-turbo | 2022 | 175 B[a] | – | – | RLHF[b] | S-FT and moderation |
| GPT-4 v.1 | 2023 | – | – | – | RLHF[b] | S-RLHF, RBRMs and moderation |
| GPT-4 v.2 | 2023 | – | – | – | RLHF[b] | S-RLHF, RBRMs and moderation |
| LLaMA-7b | 2023 | 6.7 B | 1.0 T | $4.02 \times 10^{22}$ | None | None |
| LLaMA-13b | 2023 | 13 B | 1.0 T | $4.55 \times 10^{22}$ | None | None |
| LLaMA-33b | 2023 | 32.5 B | 1.4 T | $2.73 \times 10^{23}$ | None | None |
| LLaMA-65b | 2023 | 65.2 B | 1.4 T | $5.50 \times 10^{23}$ | None | None |
| LLaMA-2-7b | 2023 | 7 B | 2.0 T | $8.40 \times 10^{22}$ | None | None |
| LLaMA-2-13b | 2023 | 13 B | 2.0 T | $1.60 \times 10^{23}$ | None | None |
| LLaMA-2-70b | 2023 | 70 B | 2.0 T | $8.10 \times 10^{23}$ | None | None |
| LLaMA-2-7b-chat | 2023 | 7 B | 2.0 T | $8.40 \times 10^{22}$ | RLHF (PPO and RS FT) | Supervised S-FT, S-RLHF and S-CD |
| LLaMA-2-13b-chat | 2023 | 13 B | 2.0 T | $1.60 \times 10^{23}$ | RLHF (PPO and RS FT) | Supervised S-FT, S-RLHF and S-CD |
| LLaMA-2-70b-chat | 2023 | 70 B | 2.0 T | $8.10 \times 10^{23}$ | RLHF (PPO and RS FT) | Supervised S-FT, S-RLHF and S-CD |
| BLOOM-560m | 2022 | 559 M | 350 B | $1.83 \times 10^{21}$ | None | None |
| BLOOM-1b1 | 2022 | 1.07 B | 350 B | $3.60 \times 10^{21}$ | None | None |
| BLOOM-1b7 | 2022 | 1.72 B | 350 B | $5.57 \times 10^{21}$ | None | None |
| BLOOM-3b | 2022 | 3.00 B | 350 B | $9.83 \times 10^{21}$ | None | None |
| BLOOM-7b | 2022 | 7.07 B | 350 B | $2.32 \times 10^{22}$ | None | None |
| BLOOM-176b | 2022 | 176.25 B | 366 B | $5.77 \times 10^{23}$ | None | None |
| BLOOMz-560m | 2022 | 559 M | 353.67 B | $1.87 \times 10^{21}$ | Multitask FT | None |
| BLOOMz-1b1 | 2022 | 1.07 B | 350.5 B | $3.69 \times 10^{21}$ | Multitask FT | None |
| BLOOMz-1b7 | 2022 | 1.72 B | 358.4 B | $5.70 \times 10^{21}$ | Multitask FT | None |
| BLOOMz-3b | 2022 | 3.00 B | 358.4 B | $1.00 \times 10^{22}$ | Multitask FT | None |
| BLOOMz-7b | 2022 | 7.07 B | 354.2 B | $2.38 \times 10^{22}$ | Multitask FT | None |
| BLOOMz-176b | 2022 | 176.25 B | 368 B | $5.91 \times 10^{23}$ | Multitask FT | None |

Key abbreviations include the following: FeedME, a supervised fine-tuning method using human-written demonstrations and top-quality model samples; PPO, a reinforcement learning approach with reward models trained through human comparisons; RBRMs, rule-based reward models that enhance a GPT-4 policy model by promoting safe content generation and discouraging harmful outputs during RLHF fine tuning; FT, fine tuning; CD, context distillation; and RS, rejection sampling. 'S-' indicates the method incorporates safety alignment. The sources of the specifications in the table are available in Supplementary Note 1.

[a]It is understood that GPT-3.5-turbo is an improvement on text-davinci-003 as described in ref. 44 and thus should have 175B parameters, but this information is not explicitly declared by OpenAI in any official sources.

[b]Specific method is unknown.

the same time. Table 1 summarizes the details of models in these three families. Scaling (increasing the number of parameters, data size and compute) has been identified as a key predictor for overall performance[1], and shaping (modifying the trained systems) has improved their instructability and alignment. This creates two categories. The first includes the 'raw' models—GPT-3 ada, babbage, curie and davinci—the non-chat LLaMA models and the base (non-*z*) BLOOM models. The second comprises the shaped-up models (or instruct or chat models), which incorporate some kind of instruction adaptation[22], fine tuning or safety moderation of the outputs. For our analysis, it is convenient that BLOOM and LLaMA have six and three exactly paired versions, respectively, of raw and shaped-up models to disentangle scaling up from shaping up.

Figure 1 represents how some key indicators show that the shaped-up models (in blue) are more stable to prompt variation and are more

correct, at the cost of being less concordant with human difficulty, and having more overall failures (less prudent). The indicators summarize the behaviour of five carefully selected benchmarks in the domains of simple numeracy ('addition'), vocabulary reshuffle ('anagram'), geographical knowledge ('locality'), diverse scientific skills ('science') and information-centric transformations ('transforms'). This covers a range of domains and degrees of open-endedness of the answers.

We identify good intrinsic proxies for human difficulty based on relevant literature in the first two domains ('addition' and 'anagram'), or by identifying demand-related features in the rest (excluding 'science', for which multiple human difficulty assessments were already available for all the instances[29]). To determine their quality, we conducted an extensive human study (S1) to assess which difficulty proxies best matched human expectations, and calibrate the proxies to a normalized difficulty score, ranging from 0 to 100, representing the anticipated

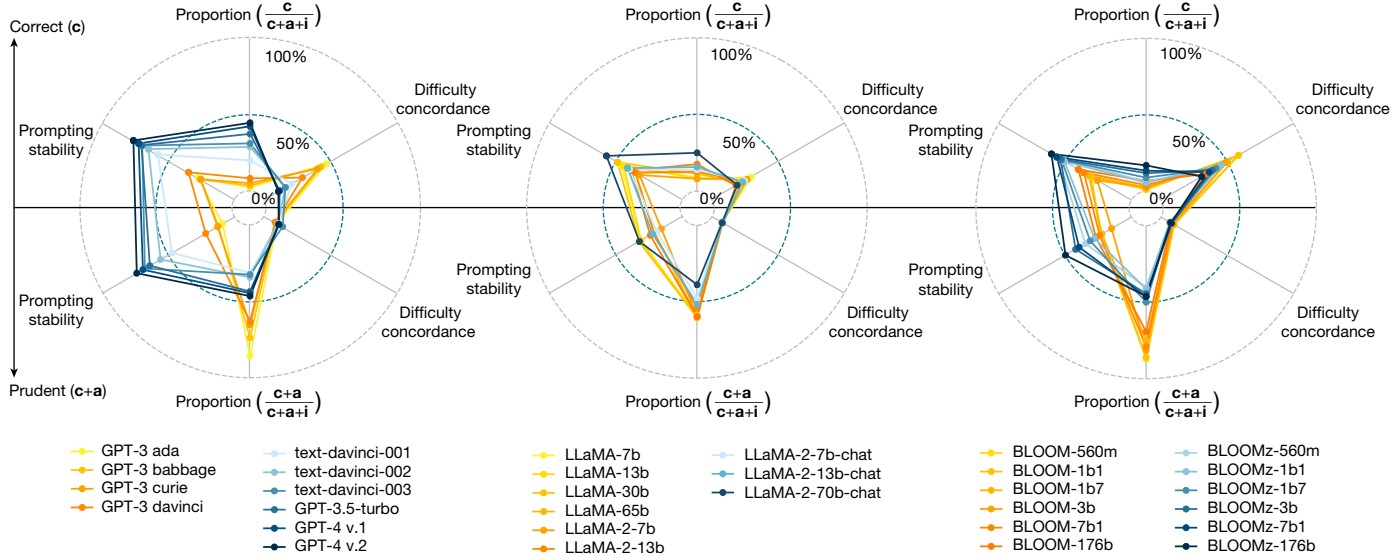

| | | |
|---|---|---|
| ● GPT-3 ada | ● text-davinci-001 | ● LLaMA-7b |
| ● GPT-3 babbage | ● text-davinci-002 | ● LLaMA-13b |
| ● GPT-3 curie | ● text-davinci-003 | ● LLaMA-30b |
| ● GPT-3 davinci | ● GPT-3.5-turbo | ● LLaMA-65b |
| | ● GPT-4 v.1 | ● LLaMA-2-7b |
| | ● GPT-4 v.2 | ● LLaMA-2-13b |
| | | ● LLaMA-2-70b |

| | | |
|---|---|---|
| ● LLaMA-2-7b-chat | ● BLOOM-560m | ● BLOOMz-560m |
| ● LLaMA-2-13b-chat | ● BLOOM-1b1 | ● BLOOMz-1b1 |
| ● LLaMA-2-70b-chat | ● BLOOM-1b7 | ● BLOOMz-1b7 |
| | ● BLOOM-3b | ● BLOOMz-3b |
| | ● BLOOM-7b1 | ● BLOOMz-7b1 |
| | ● BLOOM-176b | ● BLOOMz-176b |

**Fig. 1 | Key indicators for several models in GPT (OpenAI), LLaMA (Meta) and BLOOM (BigScience) families.** The raw models (yellow to orange) and the shaped-up models (light to dark blue) cluster differently. As the answers for all these models fall into three categories (correct, avoidant and incorrect), shortened as **c**, **a** and **i**, respectively, we have indicators for correctness versus avoidance + incorrectness, and prudence (correctness + avoidance) versus incorrectness. Looking at the correctness indicators (top half), which represent accurate responses, we see that the shaped-up models are more stable to prompt variations and are more frequently correct (higher correctness proportion) but are less concordant with human difficulty than the raw counterparts. Looking at the prudence indicators (bottom half), we see that the shaped-up models are also more stable to prompt variations, but fail more frequently (lower prudence proportion, by avoiding less) and are not much more concordant with human difficulty. Focusing only on the shaped-up models (in blue), we observe that the most powerful GPT-4 v.2, LLaMA-2-70b-chat and BLOOMz-176b models perform best in correctness proportion and prompting stability (top and bottom), but equal to or worse than other models for all the other indicators, with many fluctuations that do not indicate a clear positive trend in these other dimensions. Details of the indicators and data used for this plot are found in the Methods. Extended Data Table 1 provides a more detailed perspective on the same results.

percentage of failure for the 'average human'. Systematically controlling for human difficulty is crucial for the understanding of user-driven reliability: human expectations of success depend on the perception of the difficulty of instances[30–32]. Table 2 provides an overview of the five benchmarks, the intrinsic difficulty function used as a proxy for human difficulty (discussed in the Methods), some examples and the calibrated human difficulty values for the given examples.

Another necessary and innovative element in our analysis is that we consider three categories for the responses: correct, incorrect and avoidant, denoted by **c**, **i** and **a**, respectively. Avoidance in human participants has been extensively explored in psychology[33–35]. Such avoidant behaviours include procrastination, deviation, making excuses or simply not answering. For LLMs, avoidance is also referred to as hedging, refusal[3] or evasiveness[21], including fortuitous utterances or continuations that are not answers (non-conforming), and those responses at the meta-level explaining why the question is not answered (for epistemic or ethical reasons). Supplementary Table 11 shows the types of avoidance for some tasks in the five benchmarks.

Difficulty concordance, task avoidance and prompting stability must be regarded from the point of view of human users interacting with LLMs. Our human study S1 (see Supplementary Note 6) analyses whether human perceptions of difficulty in general are aligned with actual human performance and self-confidence, because this has important implications in the tasks humans decide to delegate to language models and their prompt formulation. But as crucial as the inputs are, so is the way the outputs from the model are used, verified or supervised. The context of use of both input and output determines how reliable the use of these systems is. We conducted a second human study S2 (see Supplementary Note 7), in which we explore whether human participants can accurately assess the outputs of models and thus compensate for different types of error. With a three-valued confusion matrix with correctness, avoidance and incorrectness, we can

focus on the frequency of non-avoidant cases for which humans believe the output is correct but it is not (Fig. 3).

With this setup, we investigate three core and intertwined elements that affect the reliability of LLMs from a human perspective.

1. Difficulty concordance. Are errors more likely for items that humans perceive as difficult? Do scaling and shaping eliminate errors for easy items, thereby creating areas of reliable operation?
2. Task avoidance. How often do language models give plausible but wrong answers instead of safely avoiding answering questions? Are scaled-up, shaped-up models better at avoiding errors or making them detectable for humans?
3. Prompting stability. How are correctness and avoidance affected by tangential changes in the prompt? Are scaled-up, shaped-up models less sensitive to prompt variation across difficulty levels?

We will answer these questions by using human difficulty metrics for each benchmark (see Table 2), examining different kinds of avoidance (Supplementary Table 11), and using 15 natural prompt variations— prompts conceived as genuine instructions or questions provided by humans—per benchmark (Supplementary Tables 1 and 2). Difficulty, avoidance and prompting, as well as their evolution, have been analysed from different perspectives[17–19,36–39] (see Supplementary Note 13 for a full discussion). Here we focus on the systemic interaction of these three elements from the perspective of LLM scaling and shaping up.

## Results

Figure 2 shows the results of a selection of models in the GPT and LLaMA families, increasingly scaled up, with the shaped-up models on the right, for the five domains: 'addition', 'anagram', 'locality', 'science' and 'transforms'. We see that the percentage of correct responses increases for scaled-up, shaped-up models, as we approach the last column. This

## Table 2 | Five benchmarks

| Benchmark | Examples | Cal. diff. |
|---|---|---|
| **Addition**: single-task benchmark. Arithmetic operations ranging from 1- to 100-digit additions. Difficulty: no. of carrying operations ($f_{cry}$) | Make the addition of 24427 and 7120. | 35.25 |
| | The sum of 47309068053 and 95464 is | 65.04 |
| | 18936030103235016 38430 + 98832380858765261900 | 98.67 |
| **Anagram**: single-task benchmark. Jumbled words consisting of 3 to 20 letters to be unscrambled to form meaningful words. Difficulty: no. of letters of the anagram ($f_{let}$) | Unscramble this string of letters, 'efe', to form a word. | 18.42 |
| | Rearrange the letters 'ngiotuq' to make a single word. | 50.42 |
| | Rearrange the following anagram into an English word: 'elmtweoascnednkg'. | 96.78 |
| **Locality**: single-task benchmark. Geographical knowledge about the location and size of cities relative to each other. Difficulty: Inverse of city popularity ($f_{pop}$) | Which city that is less than 27 km away from Toronto has the largest number of people? | 91.66 |
| | What is the name of the largest city (by population) that is less than 98 km away from Altea? | 92.64 |
| | Name the most populated city that is less than 39 km away from Akil. | 99.87 |
| **Science**: multitask benchmark. Basic science-related world knowledge questions and graduate-level questions in biology, physics and chemistry. Difficulty: Anticipated human difficulty ($f_{hum}$) | Definition: In this task, you need to provide the correct option for a given problem from the provided options.\nProblem: shining a light through a diamond can \nA) make a lot of bright lights shine\nB) summon a brilliant wave of colour\nC) heat up a room\nD) make a lot of money\nOutput: | 37.02 |
| | A light beam is propagating through a glass with index of refractionn. The glass is moving at constant velocity v in the same direction as the beam and toward the observer in laboratory. What is the speed of light in glass relative to the observer in laboratory? Take the speed of light in vacuum c=1.\nA. (1+n*v)/(n+v)\n B. (1-n*v)/(n+v)\n C. 1 D. (1+n*v)/(n-v)\nWith respect to the choices above, the correct one is | 71.83 |
| | Answer the following questions based on the list of available choices: \nIdentify the missing reagents in the following reaction.\n(3r,5r,7r)-adamantane-1-carboxylic acid + A -> (3r,5r,7r)-adamantane-1-carbonyl azide + B —> (3s,5s,7s)-adamantan-1-amine.\nA: A = NaN3 and B = HCl aq, Heat\nB: A = PCl5 and B = H3O+, Heat\nC: A = diphenylphosphoryl azide (DPPA) and B = H3O+, Heat\nD: A = diphenylphosphoryl azide (DPPA) and B = NaN3\nAnswer: | 99.97 |
| **Transforms**: multitask benchmark. Information-centric transformation tasks. Difficulty: Combination of input+output word count and Levenshtein distance ($f_{w+l}$) | Be concise in your answer, placed between double quotes. Do not generate any explanation or anything else apart from the requested output. Given\?double07@MI6.gov.uk'\nModify the input to display the domain of the email address of the form USER@DOMAIN. | 39.49 |
| | Consider the INPUT: \n\8:30h - Accreditation (badges)\n9:00h - Opening\n9:15h - Keynote\n10:15h - Coffee break\n10:45h - Invited Talks\n11:55h - Lightning talks\n12:05h - Panel\n13:00h - Lunch break (in the hall)\n14:30h - Keynote\n15:30h - Minibreak\n15:40h - Invited Talks\n16:50h - Panel\n17:45h - Closing remarks\\nI'd like the agenda to show a 15-minute reduction in each keynote speaker's segment, shifting the schedule to finish earlier. \nBe concise in your answer, placed between double quotes. Do not generate any explanation or anything else apart from the requested output. | 55.22 |
| | Michael Vaughn, a 63-year-old retired naval officer, presents an extensively complex medical history complicated by a litany of allergies. He battles chronic pain stemming from neuropathy for which he takes Pregabalin (Lyrica) 150 mg twice daily. Due to advanced rheumatoid arthritis, he relies on Etanercept (Enbrel) 50 mg, administered weekly via subcutaneous injection, but cannot be prescribed common NSAIDs like Ibuprofen or Naproxen due to gastrointestinal bleeding and a reported severe allergy to Aspirin (anaphylaxis). His Type 2 diabetes is managed with Insulin Aspart (NovoLog) administered via an insulin pump with doses varying according to his blood glucose readings; he experienced a life-threatening lactic acidosis episode with Metformin.\n I'd like the list of drugs that are prescribed to the patient to be arranged alphabetically and without repetitions, in the form of a clean, comma-separated list. Be concise in your answer, placed between double quotes. Do not generate any explanation or anything else apart from the requested output. | 64.76 |

Examples of each benchmark and their chosen difficulty metric are shown with their calibrated difficulty values (cal. diff.) according to human expectations.

is an expected result and holds consistently for the rest of the models, shown in Extended Data Fig. 1 (GPT), Extended Data Fig. 2 (LLaMA) and Supplementary Fig. 14 (BLOOM family).

Let us focus on the evolution of correctness with respect to difficulty. For 'addition', we use the number of carry operations in the sum ($f_{cry}$). For 'anagram', we use the number of letters of the given anagram ($f_{let}$). For 'locality', we use the inverse of city popularity ($f_{pop}$). For 'science', we use human difficulty ($f_{hum}$) directly. For 'transforms', we use a combination of input and output word counts and Levenshtein distance ($f_{w+l}$) (Table 2). As we discuss in the Methods, these are chosen as good proxies of human expectations about what is hard or easy according to human study S1 (see Supplementary Note 6). As the difficulty increases, correctness noticeably decreases for all the models. To confirm this, Supplementary Table 8 shows the correlations between correctness and the proxies for human difficulty. Except for BLOOM for addition, all of them are high.

However, despite the predictive power of human difficulty metrics for correctness, full reliability is not even achieved at very low difficulty levels. Although the models can solve highly challenging instances, they also still fail at very simple ones. This is especially evident for

'anagram' (GPT), 'science' (LLaMA) and 'locality' and 'transforms' (GPT and LLaMA), proving the presence of a difficulty discordance phenomenon. The discordance is observed across all the LLMs, with no apparent improvement through the strategies of scaling up and shaping up, confirmed by the aggregated metric shown in Fig. 1. This is especially the case for GPT-4, compared with its predecessor GPT-3.5-turbo, primarily increasing performance on instances of medium or high difficulty with no clear improvement for easy tasks. For the LLaMA family, no model achieves 60% correctness at the simplest difficulty level (discounting 25% random guess for 'science'). The only exception is a region with low difficulty for 'science' with GPT-4, with almost perfect results up to medium difficulty levels.

Focusing on the trend across models, we also see something more: the percentage of incorrect results increases markedly from the raw to the shaped-up models, as a consequence of substantially reducing avoidance (which almost disappears for GPT-4). Where the raw models tend to give non-conforming outputs that cannot be interpreted as an answer (Supplementary Fig. 16), shaped-up models instead give seemingly plausible but wrong answers. More concretely, the area of avoidance in Fig. 2 decreases drastically from GPT-3 ada to text-davinci-003

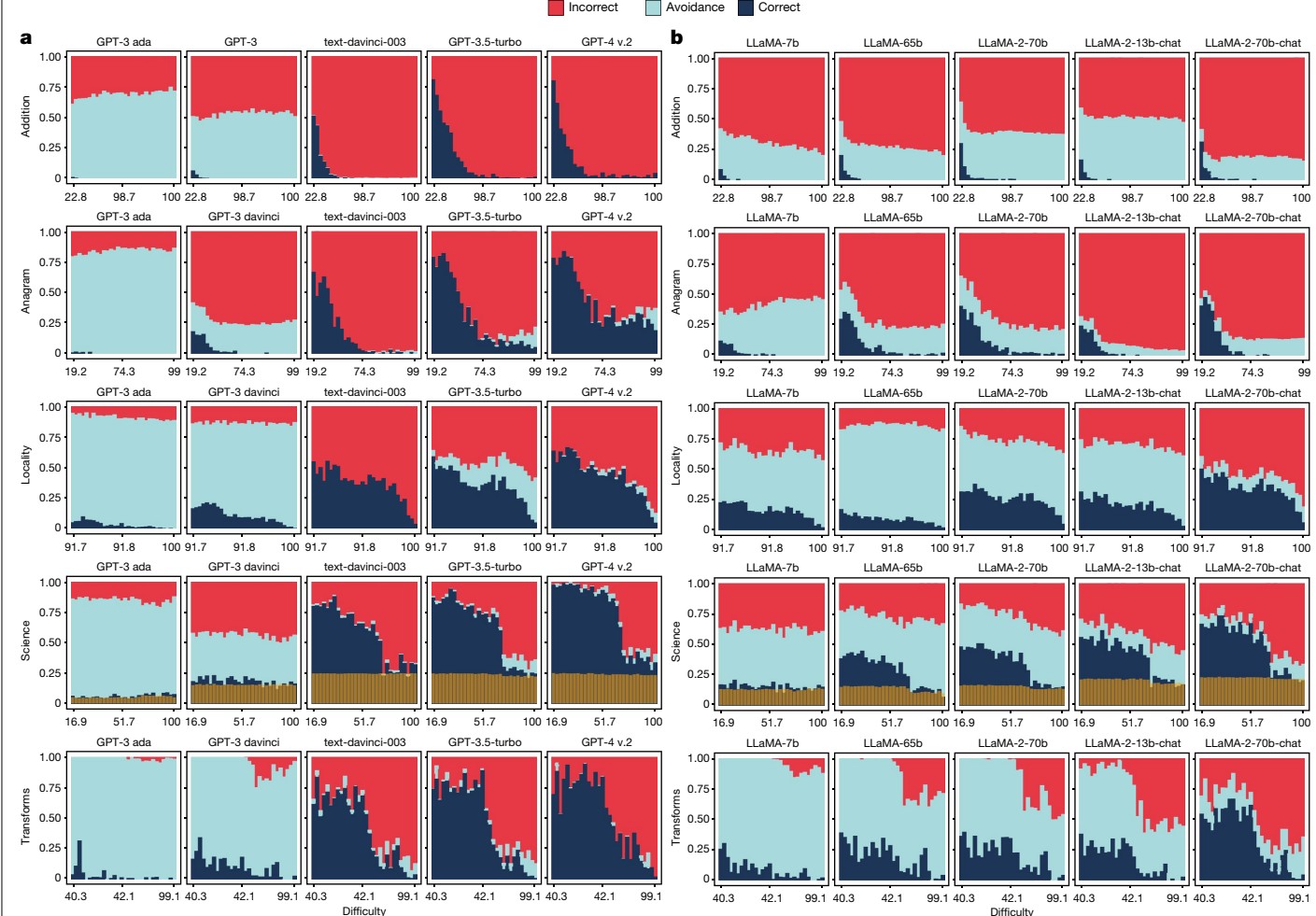

**Fig. 2 | Performance of a selection of GPT and LLaMA models with increasing difficulty.** The values are split by correct, avoidant and incorrect results. For each combination of model and benchmark, the result is the average of 15 prompt templates (see Supplementary Tables 1 and 2). For each benchmark, we show its chosen intrinsic difficulty, monotonically calibrated to human expectations on the $x$ axis for ease of comparison between benchmarks.

The $x$ axis is split into 30 equal-sized bins, for which the ranges must be taken as indicative of different distributions of perceived human difficulty across benchmarks. For 'science', the transparent yellow bars at the bottom represent the random guess probability (25% of the non-avoidance answers). Plots for all GPT and LLaMA models are provided in Extended Data Figs. 1 and 2 and for the BLOOM family in Supplementary Fig. 14.

and is replaced with increasingly more incorrect answers. Then, for GPT-3.5-turbo, avoidance increases slightly, only to taper off again with GPT-4. This change from avoidant to incorrect answers is less pronounced for the LLaMA family, but still clear when comparing the first with the last models. This is summarized by the prudence indicators in Fig. 1, showing that the shaped-up models perform worse in terms of avoidance. This does not match the expectation that more recent LLMs would more successfully avoid answering outside their operating range. In our analysis of the types of avoidance (see Supplementary Note 15), we do see non-conforming avoidance changing to epistemic avoidance for shaped-up models, which is a positive trend. But the pattern is not consistent, and cannot compensate for the general drop in avoidance.

Looking at the trend over difficulty, the important question is whether avoidance increases for more difficult instances, as would be appropriate for the corresponding lower level of correctness. Figure 2 shows that this is not the case. There are only a few pockets of correlation and the correlations are weak. This is the case for the last three GPT models for 'anagram', 'locality' and 'science' and a few LLaMA models for 'anagram' and 'science'. In some other cases, we see an initial increase in avoidance but then stagnation at higher difficulty levels. The percentage of avoidant answers rarely rises quicker than the

percentage of incorrect ones. The reading is clear: errors still become more frequent. This represents an involution in reliability: there is no difficulty range for which errors are improbable, either because the questions are so easy that the model never fails or because they are so difficult that the model always avoids giving an answer.

We next wondered whether it is possible that this lack of reliability may be motivated by some prompts being especially poor or brittle, and whether we could find a secure region for those particular prompts. We analyse prompt sensitivity disaggregating by correctness, avoidance and incorrectness, using the prompts in Supplementary Tables 1 and 2. A direct disaggregation can be found in Supplementary Fig. 1, showing that shaped-up models are, in general, less sensitive to prompt variation. But if we look at the evolution against difficulty, as shown in Extended Data Figs. 3 and 4 for the most representative models of the GPT and LLaMA families, respectively (all models are shown in Supplementary Figs. 12, 13 and 15), we observe a big difference between the raw models (represented by GPT-3 davinci) and other models of the GPT family, whereas the LLaMA family underwent a more timid transformation. The raw GPT and all the LLaMA models are highly sensitive to the prompts, even in the case of highly unambiguous tasks such as 'addition'. Difficulty does not seem to affect sensitivity very much, and for easy instances, we see that the raw models (particularly,

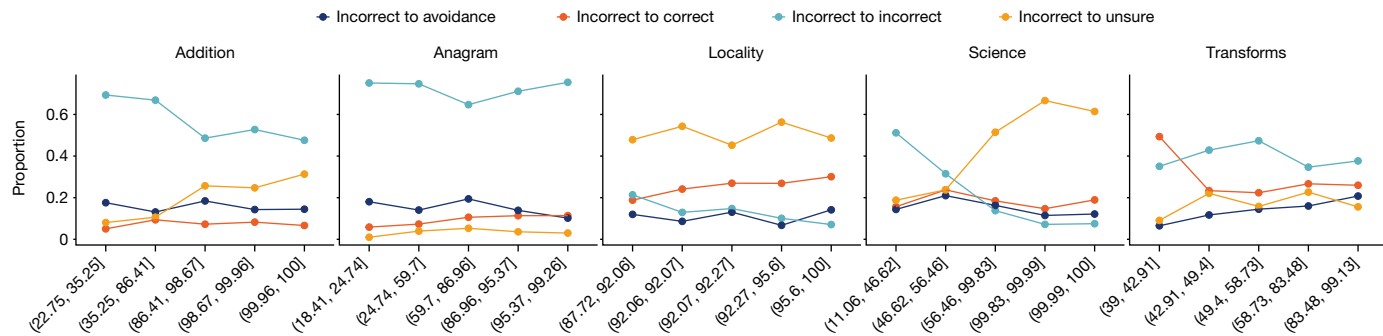

**Fig. 3 | Evolution of types of supervision error versus difficulty according to human survey S2.** In the survey (Supplementary Fig. 4), participants have to determine whether the output of a model is correct, avoidant or incorrect (or do not know, represented by the 'unsure' option in the questionnaire). Difficulty ($x$ axis) is shown in equal-sized bins. We see very few areas where the dangerous error (incorrect being considered correct by participants) is sufficiently low to consider a safe operating region.

GPT-3 davinci and non-chat LLaMA models) have some capacity that is unlocked only by carefully chosen prompts. Things change substantially for the shaped-up models, the last six GPT models and the last three LLaMA (chat) models, which are more stable, but with pockets of variability across difficulty levels.

Overall, these different levels of prompt sensitivity across difficulty levels have important implications for users, especially as human study S2 shows that supervision is not able to compensate for this unreliability (Fig. 3). Looking at the correct-to-incorrect type of error in Fig. 3 (red), if the user expectations on difficulty were aligned with model results, we should have fewer cases on the left area of the curve (easy instances), and those should be better verified by humans. This would lead to a safe haven or operating area for those instances that are regarded as easy by humans, with low error from the model and low supervision error from the human using the response from the model. However, unfortunately, this happens only for easy additions and for a wider range of anagrams, because verification is generally straightforward for these two datasets.

Our observations about GPT and LLaMA also apply to the BLOOM family (Supplementary Note 11). To disentangle the effects of scaling and shaping, we conduct an ablation study using LLaMA and BLOOM models in their shaped-up versions (named chat and z, respectively) and the raw versions, with the advantage that each pair has equal pre-training data and configuration. We also include all other models with known compute, such as the non-instruct GPT models. We take the same data summarized in Fig. 1 (Extended Data Table 1) and perform a scaling analysis using the FLOPs (floating-point operations) column in Table 1. FLOPs information usually captures both data and parameter count if models are well dimensioned[40]. We separate the trends between raw and shaped-up models. The fact that correctness increases with scale has been systematically shown in the literature of scaling laws[1,40]. With our data and three-outcome labelling, we can now analyse the unexplored evolution of avoidance and incorrectness (Fig. 4, left).

As evident in Fig. 4, avoidance is clearly much lower for shaped-up models (blue) than for raw models (orange), but incorrectness is much higher. But even if correctness increases with scale, incorrectness does not decrease; for the raw models, it increases considerably. This is surprising, and it becomes more evident when we analyse the percentage of incorrect responses for those that are not correct in ($\mathbf{i}$/($\mathbf{a} + \mathbf{i}$) in our notation; Fig. 4 (right)). We see a large increase in the proportion of errors, with models becoming more ultracrepidarian (increasingly giving a non-avoidant answer when they do not know, consequently failing proportionally more).

We can now take all these observations and trends into account, in tandem with the expectations of a regular human user (study S1) and the limited human capability for verification and supervision (study S2). This leads to a re-understanding of the reliability evolution

of LLMs, organized in groups of two findings for difficulty discordance (F1$_a$ and F1$_b$), task avoidance (F2$_a$ and F2$_b$) and prompt sensitivity (F3$_a$ and F3$_b$):

F1$_a$—human difficulty proxies serve as valuable predictors for LLM correctness. Proxies of human difficulty are negatively correlated with correctness, implying that for a given task, humans themselves can have approximate expectations for the correctness of an instance. Relevance: this predictability is crucial as alternative success estimators when model self-confidence is either not available or markedly weakened (for example, RLHF ruining calibration[3,41]).

F1$_b$—improvement happens at hard instances as problems with easy instances persist, extending the difficulty discordance. Current LLMs clearly lack easy operating areas with no error. In fact, the latest models of all the families are not securing any reliable operating area. Relevance: this is especially concerning in applications that demand the identification of operating conditions with high reliability.

F2$_a$—scaling and shaping currently exchange avoidance for more incorrectness. The level of avoidance depends on the model version used, and in some cases, it vanishes entirely, with incorrectness taking important proportions of the waning avoidance (that is, ultracrepidarianism). Relevance: this elimination of the buffer of avoidance (intentionally or not) may lead users to initially overtrust tasks they do not command, but may cause them to be let down in the long term.

F2$_b$—avoidance does not increase with difficulty, and rejections by human supervision do not either. Model errors increase with difficulty, but avoidance does not. Users can recognize these high-difficulty instances but still make frequent incorrect-to-correct supervision errors. Relevance: users do not sufficiently use their expectations on difficulty to compensate for increasing error rates in high-difficulty regions, indicating over-reliance.

F3$_a$—scaling up and shaping up may not free users from prompt engineering. Our observations indicate that there is an increase in prompting stability. However, models differ in their levels of prompt sensitivity, and this varies across difficulty levels. Relevance: users may struggle to find prompts that benefit avoidance over incorrect answers. Human supervision does not fix these errors.

F3$_b$—improvement in prompt performance is not monotonic across difficulty levels. Some prompts do not follow the monotonic trend of the average, are less conforming with the difficulty metric and have fewer errors for hard instances. Relevance: this non-monotonicity is problematic because users may be swayed by prompts that work well for difficult instances but simultaneously get more incorrect responses for the easy instances.

As shown in Fig. 1, we can revisit the summarized indicators of the three families. Looking at the two main clusters and the worse results of the shaped-up models on errors and difficulty concordance, we may rush to conclude that all kinds of scaling up and shaping up are

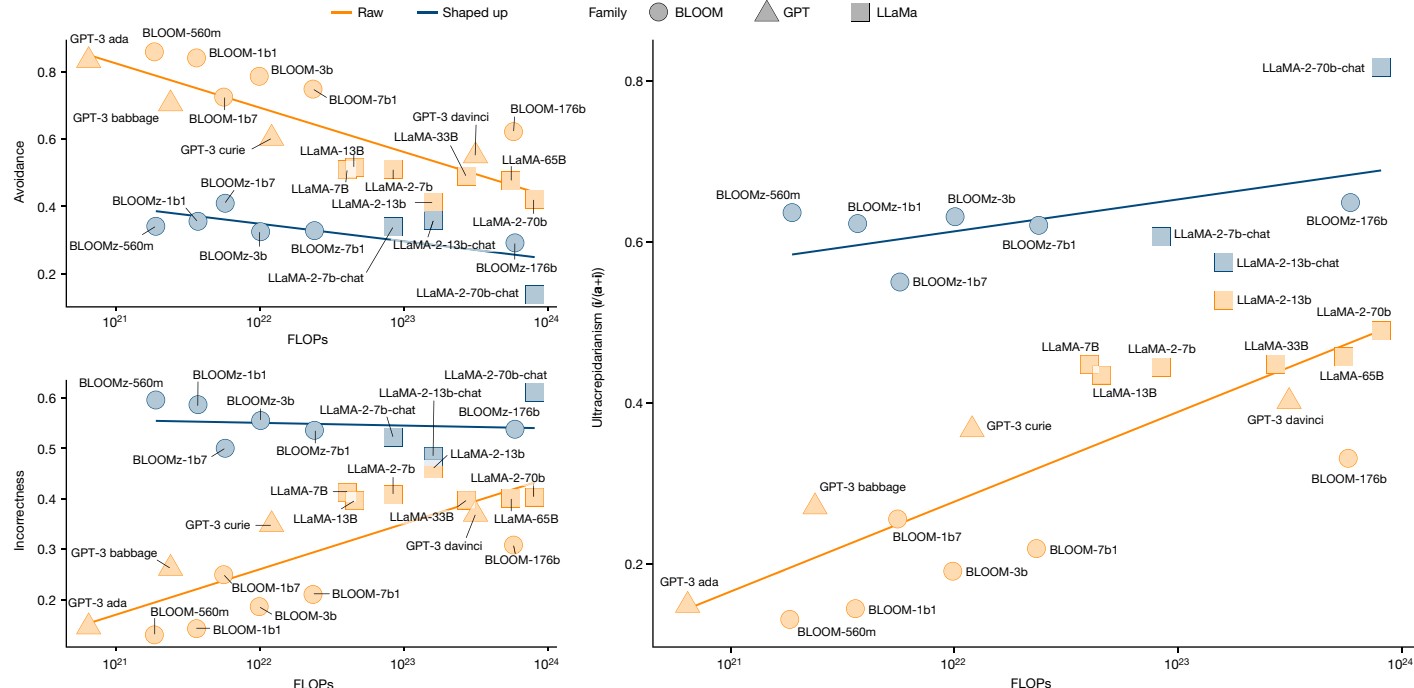

**Fig. 4 | Scaling analysis of LLaMA and BLOOM families and non-instruct GPT models.** The plot uses a logarithmic scale for FLOPs. The focus is on avoidance (**a**; top left), incorrectness (**i**; bottom left) and ultracrepidarianism ($i/(a+i)$; right)—the proportion of incorrect over both avoidant and incorrect answers.

inappropriate for ensuring user-driven reliability in the future. However, these effects may well be the result of the specific aspirations for these models: higher correctness rates (to excel in the benchmarks by getting more instances right but not necessarily all the easy ones) and higher instructability (to look diligent by saying something meaningful at the cost of being wrong). For instance, in scaling up, there is a tendency to include larger training corpora[42] with more difficult examples, or giving more weight to authoritative sources, which may include more sophisticated examples[43], dominating the loss over more straightforward examples. Moreover, shaping up has usually penalized answers that hedge or look uncertain[3]. That makes us wonder whether this could all be different.

## Discussion

In this paper, we have conducted two human studies. The first investigates perceived and actual difficulty for participants to respond to an input (to determine whether difficulty expectations are correlated with difficulty proxies). The second includes participants supervising or verifying the output of a model (to determine whether humans will take incorrect responses as correct). Maximizing difficulty concordance and reducing possible incorrect-to-correct errors in human verification could be introduced in the loss function when training and shaping up these models. For this, collective efforts are needed to build larger datasets of human difficulty expectations and output supervision. With these data, more qualified than traditional human feedback, AI itself can be used to train supervisors that perform this shaping up, provided the aim is not to eliminate evasiveness as in ref. 21, but to find the right level of avoidance. Specialized language models in medicine and other critical areas may be designed with reject options, or coupled with external AI supervisors, thereby favouring avoidance by teaching the AI models when to refrain from answering[37]. These interventions should make LLMs exhibit enhanced human-like and human-aligned characteristics that ensure reliability. Until this is done, and given the high penetration of LLM use in the general population, we raise awareness that relying on human oversight for

these systems is a hazard, especially for areas for which the truth is critical.

Finally, we include some limitations of our analysis and the future work that emanates from them. The first limitation of our study lies in the recruitment of participants who are mostly non-experts. We have to take this into account when interpreting the calibrated difficulty values, which are usually high for some benchmarks, as there is a high number of questions that cannot be solved by the general population. However, our motivation was to capture the same human population to estimate expected instance difficulties that are comparable across all the datasets. A second limitation is that our sample of 'natural' prompts was collected from a diversity of sources, but we did not have access to the frequency in which a prompt may appear in a real scenario. Last, we have only covered a sample of families with specific trajectories, excluding LLMs that delegate tasks to external tools or use sophisticated reasoning techniques, which may show different dynamics. The GPT family has been at the forefront in performance and has been used over a few years, making OpenAI extremely influential in the development of other language models[22,23]. In fact, the OpenAI application programming interface has the most dependencies when the ecosystems of foundation models are analysed[24]. LLaMA and BLOOM have a more open and systematic lineup of models, not only allowing for the disentanglement between scaling and shaping but also paving the way for an incremental analysis of their evolution using our methodology and code, in the fast-changing context of LLM development. Highlighting the reliability issues of these families and introducing new abstractions and tools for analysis is of utmost importance, enabling other researchers to explore different pathways for the scaled-up, shaped-up models of the future.

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

## Methods

We now explain our choices of benchmarks, prompt templates, difficulty functions, response scoring, general experimental design and the key metrics used to evaluate the models.

### Benchmarks and factors of difficulty

For the generality of our analysis, we selected five distinct benchmarks to reduce confounding factors as much as possible: simple numeracy ('addition'), vocabulary reshuffle ('anagram'), geographical knowledge ('locality'), basic and advanced science questions ('science') and information-centric transformations ('transforms'). These represent core skills (numerical, linguistic and knowledge) and more diverse ecologically valid scenarios, with some of them having extremely simple formulations and others requiring deep understanding of the information presented, as well as the integration of data from multiple sources. Closed-ended questions are typical of LLM research[3], such as those found in the 'science' benchmark, but gradually more open-ended tasks ('addition', 'anagram', 'locality' and 'transforms') better represent a wider and more realistic use of LLMs.

- Addition. This benchmark involves sums, prompting the LLMs by asking for the result of adding two addends (such as '3 + 7 ='). The examples in our analysis range from 1- to 100-digit additions. Because language models can not only memorize small additions but also generalize to cope with any combination of larger digits, this task is appropriate for analysing difficulty trends. With respect to the difficulty of 'addition', the number of digits and carry operations affect human performance on addition tasks.

- Anagram. The use of anagrams as a way of assessing aspects of problem solving dates back to 1916 (ref. 45), and researchers have been using anagrams to examine a variety of phenomena, such as the cognitive processes involved in problem solving[46]. An 'anagram' task is a word puzzle in which the participant or model is presented with a jumbled string of letters, and the objective is to find a word that can be formed using all the letters given. The examples in our analysis range from 3-letter words to 20-letter words. This task involves letter manipulation and good recall from an extensive vocabulary. One peculiar element of this task is that it is easy to verify. The difficulty of anagrams is mostly influenced by the frequency of the letters and the word, the number of letters and the degree of rearrangement required.

- Locality. This benchmark contains questions relating to geographical knowledge, inspired by some cognitive models of distance estimation[47]. The examples in our analysis ask questions about the location and size of cities in relation to each other, by giving an input city and a randomly generated distance ($d$, ranging from 1 to 1,000 km). The LLM is asked to identify the most populous city (the target city) in a radius of $d$ km from the input city. This task requires geographical knowledge and reasoning. For this benchmark, potential human difficulty factors could be the city or country popularity, their population and so on.

- Science. This benchmark integrates multiple-choice questions from basic science as collected by OpenBookQA, complemented with more advanced science questions from Google-proof Q&A (GPQA). They represent tasks that LLMs are likely to encounter in educational, academic and research settings[6,8,48], some of which require considerable time to solve. The included questions are Google-proof[49]. The 'science' benchmark, thus, includes questions of varying levels of difficulty, as determined by human judgement, providing a lens through which to examine their handling of complex, data-rich tasks in specific domains.

- Transforms. This benchmark includes a comprehensive set of information-centric transformation tasks based on real-world scenarios. It focuses on domains that are most prevalent in the use of LLMs today[50], and ensure that there is a ground truth for evaluation. We integrate not only many data-formatting tasks—a well-studied area in LLMs[51]—but also new tasks about world knowledge, information retrieval, advertising, administration, coding, scheduling and retailing. The outputs for 'transforms' may require extensive elaboration of the input (hundreds of characters) to form a correct answer, which can also be hundreds of characters long. The aim was to simulate, as closely as possible, the complexity and depth of real-world questions in a controlled experimental setting. For task difficulty, given the heterogeneity, the main factors are as general as character and word counts, and the Levenshtein distance between input and output as a proxy of transformation effort.

For the previously described domains, we found intuitive human difficulty proxies, some of which have been developed in the literature. Supplementary Note 4 provides further details on the definition of difficulty metrics and the abilities behind the features used for their definition. Using the results from human study S1, we select the difficulty functions that are most correlated with human expectations (Supplementary Table 5): $f_{cry}$ for 'addition', $f_{let}$ for 'anagram', $f_{pop}$ for 'locality' and $f_{w+l}$ for 'transforms'. For 'science', we blend and calibrate the two original human metrics into one, that is, $f_{hum}$. For all the benchmarks, we normalize the original difficulty functions using a logistic mapping to a scale ranging from 0 to 100 that corresponds to the probability of human failure as estimated by humans themselves. We need to take into account that these values are an estimate (from the human sample in S1, of their expectations) and are fitted with a two-parameter logistic function; therefore, these values between 0% and 100% have to be interpreted with caution, especially for small differences (see Supplementary Note 8 for details). Nevertheless, having all the difficulty levels on the same human-expectations scale helps with the comparison of the benchmarks.

### Data collection and generation

We first describe how the examples were collected or generated, and then the 15 prompt templates that were used for each of them.

- Addition. We randomly generate 5,000 instances, in which each addend is sampled uniformly from 1 to 100 digits. We then remove those instances for which $f_{hrm} > 50$ to prevent instances with similar or identical numbers of digits in both addends from dominating the upper difficulty bins. This is because, for example, if the difficulty is the harmonic mean, the bins with $f_{hrm} > 90$ would be dominated by instances in which both addends have very high numbers of digits (that is, at least 82 digits). A similar phenomenon also occurs with other difficulty levels, but with the previous criterion considered, the problem is well mitigated. This results in a final sample of 3,142 instances.

- Anagram. We use the Google Web Trillion Word Corpus[52], containing the frequency of more than 300,000 most commonly used single words on the Web in English. From this corpus, we randomly choose up to 100 English words with 3–20 letters, resulting in a total of 1,570 words. There are fewer than 1,800 instances because there are fewer than 100 English words with 17–20 letters. Then, we shuffle the order of letters randomly to map these words into 1,570 anagrams. We make sure the resultant permutation is not the same as the original word.

- Locality. We use the World Cities Database[53], which provides an up-to-date database of the cities and towns globally. From this database, we first exclude cities with non-unique names across the globe. Next, we remove cities with more than one word or non-standard letters in the 26-character Latin alphabet (for example, Buenos Aires or Chŏngjin) to enhance the quality and ease of the response-scoring method. After the previous selection procedure, we seek to form a final sample that covers instances with different difficulty levels (or bins) as equally as possible. Thus, we perform binning on the difficulty function ($f_{pop}$) to produce 100 bins in which we extract up to 50 instances from each bin randomly, resulting in a total of 2,341 instances. Again, there are fewer than 5,000 instances because some bins contain fewer than 50 instances.

- **Science.** This benchmark is built by integrating multiple-choice questions from educational settings: OpenBookQA[29] and GPQA[49]. OpenBookQA is a collection of multiple-choice questions in basic science, based on 1,329 established facts. We randomly sampled 1,000 questions from OpenBookQA. To complement the benchmark with more advanced science questions, we included GPQA[49]—a dataset containing 546 graduate-level questions written by domain experts that challenge LLMs to demonstrate a deep understanding of biology, physics and chemistry. We exclude two lengthy questions that exceed the context window limit for some of the models that we analyse.

- **Transforms.** This benchmark includes a comprehensive set of information-centric transformation tasks based on real-world scenarios. We integrate many data-formatting questions from a data-wrangling dataset[51] and from a 'natural instructions' dataset[54], manually regenerating or adapting some of them. We also also introduce new tasks about world knowledge, information retrieval, advertising, administration, coding, scheduling and retailing, reflecting a wide range of real user interactions with language models. The benchmark integrates 73 different tasks, with 10 instances each, totalling 730 items.

## Prompt generation

Notably, 'addition', 'anagram', 'locality' and parts of 'transforms' are newly introduced in this work. All five benchmarks are further supplemented with human data (see Supplementary Note 5) for calibrating difficulty levels and supervision, as well as a new variable describing the human-calibrated difficulty for each data instance.

Each example in each benchmark is run through an LLM using 15 different prompts, which are the same for all the examples in the benchmark. The generation of prompt templates aims to fulfil three requirements. First, the prompts should be as natural as possible, because we try to model a situation in which humans interact with LLMs in a similar way to how they would talk to other humans. Second, these prompts should be derived from or inspired by real-world sources, except for minor variations and adaptations. Third, we need to have sufficient coverage for and diversity of prompt templates, to robustly analyse sensitivity, omitting those that are too similar. This process results in 15 natural prompt templates for each benchmark, extracted from or inspired by textbooks, scientific literature, academic exams and the internet. Supplementary Note 2 describes further details about these prompt templates and their sources.

## Response scoring

Scoring the validity of the responses of LLMs can be challenging, given that their raw text response can vary in different ways. For example, some responses are highly elaborate, whereas other responses are concise and straight to the point. Some responses are unrelated or digress from the proposed question, or are just excessively verbose, providing the answer in a larger response sequence surrounded by arbitrary information. Because our analysis uses three classes (correct, incorrect and avoidant), the confusion matrices have nine cells, making grading more challenging, and the traditional intuition and terminology of false positives, false negatives, sensitivity, specificity, precision and recall cannot be easily extended to these three-outcome situations. In Supplementary Note 13, we discuss how different groups of cells are named.

Manual scoring becomes infeasible due to the massive amount of answers we collect (approximately 4.2 million). Fortunately, despite the arbitrary responses of the models, they do exhibit a set of common patterns. We succeeded in scoring these responses using simple algorithmic conditions and regular expressions that provide great scoring accuracy (see Supplementary Note 3).

## Experimental setup

The LLMs are described in Table 1. All the models were queried with the temperature parameter set to zero and no system prompt. For local inference, we made use of a shared cluster of six nodes with 8×

NVIDIA A40 48 GB graphics processing units. All local inferences were single node, made use of the Hugging Face Transformers and Accelerate libraries, and were without quantization of the models, with the exception of BLOOMz (see below). The total compute estimate for all the experiments (including reruns and discarded results) is estimated to be about 100 compute days on a single 8× A40 node.

- **GPT:** we used ten models from the GPT family (OpenAI)[55]. The first four models, GPT-3 ada, babbage, curie and davinci, are the original raw models in the family[14]. The subsequent three are the later and more powerful model variants (the InstructGPT versions of davinci called text-davinci-001, text-davinci-002 and text-davinci-003)[5], which are shaped up by fine tuning with human feedback. The last three models are also fine-tuned with human feedback and further include a moderation post-filtering mechanism[3]. GPT-3.5-turbo was built as 'gpt-3.5-0301' (March 2023), and the two GPT-4 models differ in the time of their build ('gpt-4-0314' and 'gpt-4-0613'). All these models were accessed through the public application programming interface (API). We used the ChatCompletion API (https://platform.openai.com/docs/api-reference/chat/streaming).

- **LLaMA:** we used four different scales of the first LLaMA version[25]: 7b, 13b, 30b and 65b. For LLaMA-2 (ref. 26), there is no 30b variant available, but we used all the other sizes (7b, 13b and 70b), including the corresponding chat variants, which incorporate various shaping techniques. All the inferences were run locally, except for LLaMA-65b, for which we used the Hugging Face API, and LLaMA-2 (non-chat), for which we used the Together.AI API.

- **BLOOM:** we used the six different scales (560m to 176b) of the BLOOM[27] and BLOOMz[28] models, the latter of which was an update that added (multilingual) multitask fine tuning (also known as instruction tuning). As before, all the inferences on the small models were run locally. The biggest variant for BLOOM was run through the Hugging Face API. BLOOMz was run locally, but with NF4 quantization[56] to fit into a single node.

The number of tokens was adjusted for the benchmark: 'addition' = 256, 'anagram' = 72, 'locality' = 132, 'science'-OBQA = 72, 'science'-GPQA = 384 for all the models, except for GPT-3.5 and GPT-4, which used 1,000 tokens. For 'transforms', we used the formula round(max(72,output_length)) × 3/4. All these numbers ensured that we could get long enough responses that include the answers for approximately 99% of instances and substantially reduce the cost. We used the default values for the stopping condition and the rest of the parameters.

## Evaluation of models

For each difficulty function, we rank the data examples and separate them into 30 equal-sized bins based on their difficulty values. With this, we calculate bin-wise correctness, incorrectness and avoidance rates. Then, we plot these rates as a stacked bar chart (Fig. 2), for which we calculate the Spearman rank correlation (Supplementary Table 8). Similarly, we illustrate the prompt sensitivity of correctness, incorrectness and avoidance by plotting the performance of each individual prompt template for these dimensions across each model (Supplementary Figs. 12, 13 and 15).

Moreover, we delineate six reliability indicators for all the models in GPT (OpenAI), LLaMA (Meta) and BLOOM (BigScience) families (Fig. 1). There are three categories of answers: correct (**c**), avoidant (**a**) and incorrect (**i**). By separating correct from avoidant or incorrect (**c** vs **a** + **i**), the design or evaluation focus is put on accuracy, whatever damage the errors may do, but if correct or avoidant is placed against incorrect (**c** + **a** vs **i**), the focus is put on reliability. Instead of non-incorrect, we use the term prudent to refer to the group of correct or avoidant answers as a whole. Accounting for these groups, we have two versions for each of the following indicators.

- **Proportion:** this measures the percentage of some of the groups of responses. In particular, the correctness proportion is the

probability of giving a correct answer, that is, $\mathbb{P}(\mathbf{c}\langle j, p\rangle)$, where $j$ and $p$ refer to an instance and a prompt for that instance, respectively, and $\mathbf{c}$ represents correctness. The prudence proportion is the probability of giving a prudent (non-incorrect) answer, that is, $\mathbb{P}(\neg\mathbf{i}\langle j, p\rangle)$, where $\mathbf{i}$ represents incorrectness.

- Prompting stability: this is the probability that the answer to an instance remains in the same group after changing the prompt. Let us define $s^{\mathbf{c}}$ as $\mathbb{P}(\mathbf{c}\langle j, p'\rangle|\mathbf{c}\langle j, p\rangle)$, where $j$ refers to an instance, and $p$ and $p'$ refer to two prompts for that instance (which are not necessarily different). This measures just the probability that given an instance–prompt pair that is correct (sampling uniformly from all these positive pairs), we still get a correct answer if we sample another prompt. Similarly, we define $s^{\neg\mathbf{c}}$ as $\mathbb{P}(\neg\mathbf{c}\langle j, p'\rangle|\neg\mathbf{c}\langle j, p\rangle)$. Finally, we define correctness prompting stability as $s_{\mathbf{c}} = 0.5\,(s^{\mathbf{c}} + s^{\neg\mathbf{c}})$ and prudence prompting stability as $s_{\mathbf{p}} = 0.5\,(s^{\mathbf{i}} + s^{\neg\mathbf{i}})$. It can be shown that these metrics go between 0.5 and 1; we scale them to go from 0 to 100.
- Difficulty concordance: this measures the degree to which higher difficulty implies lower quality of results. We will use the generality metric introduced in ref. 57, as it aligns precisely with the concept of difficulty concordance. Technically, generality is a non-parametric metric that measures how much the mass of success conforms to a step function. If success were distributed like a descending logistic curve, generality would be equal to the maximum slope of a descending curve, that is, the steeper the slope, the higher the generality metric gets, and thus has a higher level of difficulty concordance. A model being good for all instances up to a given difficulty and then bad for more difficult instances would have perfect concordance. Therefore, this is not the same as correlation (see Supplementary Table 8). Again, we define two versions, namely, correctness difficulty concordance (which calculates the generality for the correct answers) and prudence difficulty concordance (which calculates the generality for the prudent (non-incorrect) answers). We transform it with $x/(x + 1) \times 100$ to get a value between 0 and 100. For 'science', we discount 25% of non-avoidant responses to account for random guesses.

We propose that researchers use these six reliability metrics for the initial analysis of the reliability of any existing or future LLM. In Fig. 1, we do this by averaging the values procured from the five benchmarks to provide a succinct summary of the reliability fluctuations of the three families (detailed data are shown in Extended Data Table 1).

Following the advice in ref. 58, we strongly recommend that these metrics are always accompanied by a detailed analysis and breakdown of results, as we have done in this paper with the other plots.

### Inclusion and ethics

The ethical committee of the Universitat Politècnica de València (UPV) approved the present work. We conducted two human studies in which we recorded the perceived and actual difficulty that participants have when solving some tasks (S1) and scoring the tasks solved by LLMs (S2). The studies were performed using surveys implemented in the Concerto platform. The users were recruited by using the Prolific platform. All participants provided written informed consent on enrolment. They received compensation at a rate of £9 per hour. In this work, we used LLMs, which are trained on very different sources of data and may have important ethical consequences, such as generating incorrect responses that look plausible. The domains used in our experiments and the examples included in the manuscript do not generate any specific ethical issue. We only use examples and prompts in English.

### Data availability

All data, including existing and newly created datasets, prompts, model responses, grading (manual and automatic) and the human study data (questions and responses) are available on Zenodo at https://doi.org/10.5281/zenodo.12794511 (ref. 59). To hinder data contamination from automated web scraping, the relevant data files are provided as a password-encrypted zip file, for which the access code is also provided in the repository. Source data are provided with this paper.

### Code availability

All code, including for data analysis, human study, plotting, algorithmic grading conditions and interacting with language models, is available on Zenodo at https://doi.org/10.5281/zenodo.12794511 (ref. 59) and on GitHub at https://github.com/wschella/llm-reliability.

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

**Acknowledgements** We thank OpenAI for granting us research access to several LLMs to conduct the experiments in this paper; Meta for giving us access to the weights of their models; BigScience initiative for open access to their LLMs; L. Sun for help in configuring Concerto, the testing platform from Cambridge Psychometrics Centre that we used for the human study; M. Tešić for giving us some suggestions for the human study; and R. Burnell for valuable feedback on some of the concepts and metrics used in this paper. We acknowledge support from the following institutions: Long-Term Future Scholarship funded by Open Philanthropy; Grant for Master Studies funded by ValgrAI—Valencian Graduate School and Research Network for Artificial Intelligence and Generalitat Valenciana; FISCALTICS (I+D+i PID2022-140110OA-I00), funded by MICIU/AEI/10.13039/501100011033 and by ERDF, EU; CIPROM/2022/6 (FASSLOW) and IDIFEDER/2021/05 (CLUSTERIA) funded by Generalitat Valenciana; the EC H2020-EU grant agreement no. 952215 (TAILOR); US DARPA HR00112120007 (RECoG-AI) and Spanish grant PID2021-122830OB-C42 (SFERA) funded by MCIN/AEI/10.13039/501100011033 and 'ERDF A way of making Europe'; and the Vic. Inv. of the Universitat Politècnica de Valencia under DOCEMPR21, DOCEMPR22 and DOCEMPR23.

**Author contributions** All authors contributed to the conception of the project, the collection of benchmarks, the prompts and the difficulty metrics, as well as the choice of model families and experimental methodology. L.Z., W.S. and Y.M.-D. ran the core experiments, with W.S. managing the infrastructure for locally running the LLMs. L.Z., Y.M.-D. and J.H.-O. scored the results semi-automatically. All authors devised the human studies, which were implemented and run by W.S. and C.F., and processed by C.F., W.S., L.Z., Y.M-D and J.H.-O. Ethical committee approvals were managed by C.F. and Y.M.-D. Result analysis and plotting were prepared by L.Z., F.M.-P. and J.H.-O., with F.M.-P. generating most of the figures. All authors edited and revised the manuscript. J.H.-O. supervised the project.

**Competing interests** The authors declare no competing interests. Some authors received economic compensation for red teaming some of the models that appear in this study, as well as for red teaming other models created by the same companies.

**Additional information**
**Correspondence and requests for materials** should be addressed to José Hernández-Orallo.

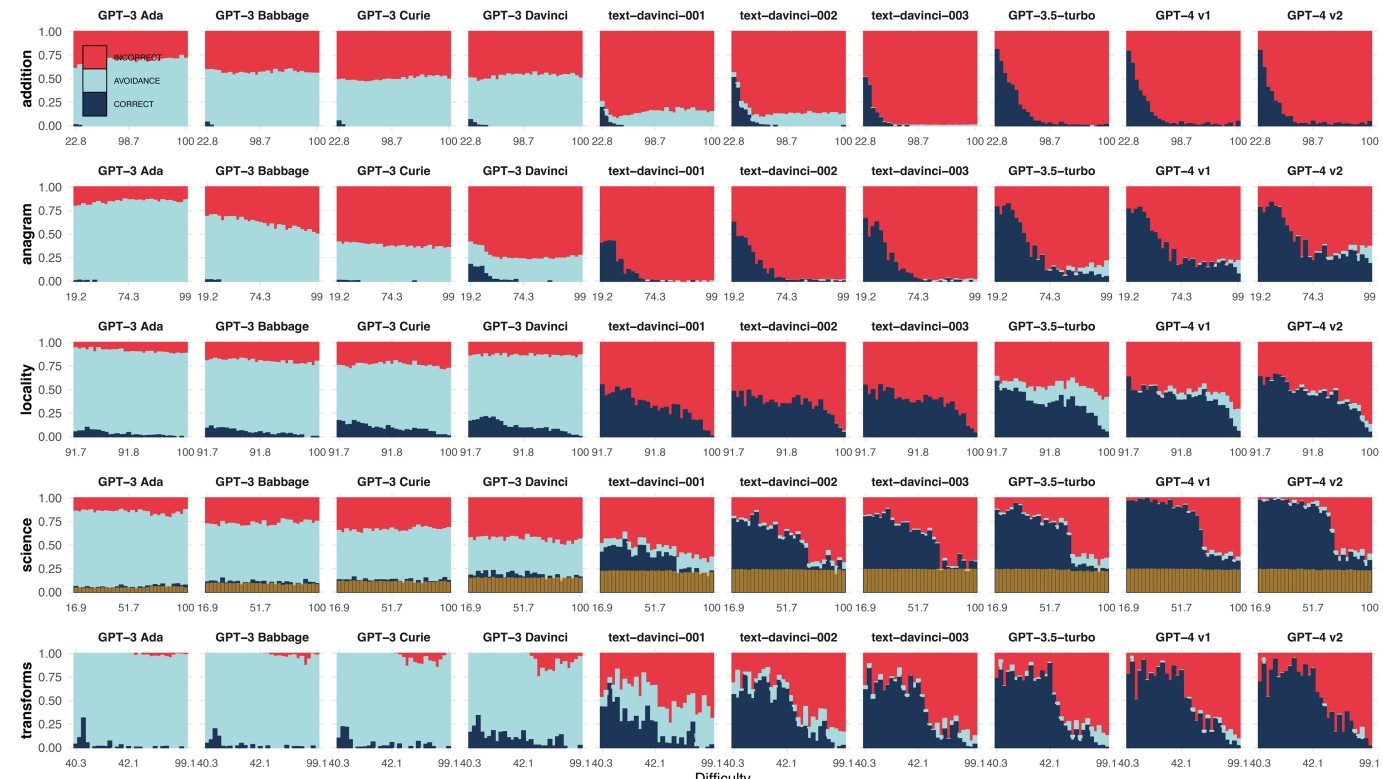

**Extended Data Fig. 1 | Performance of GPT models over difficulty.** The values are split by correct, avoidant and incorrect results. Details as in Fig. 2.

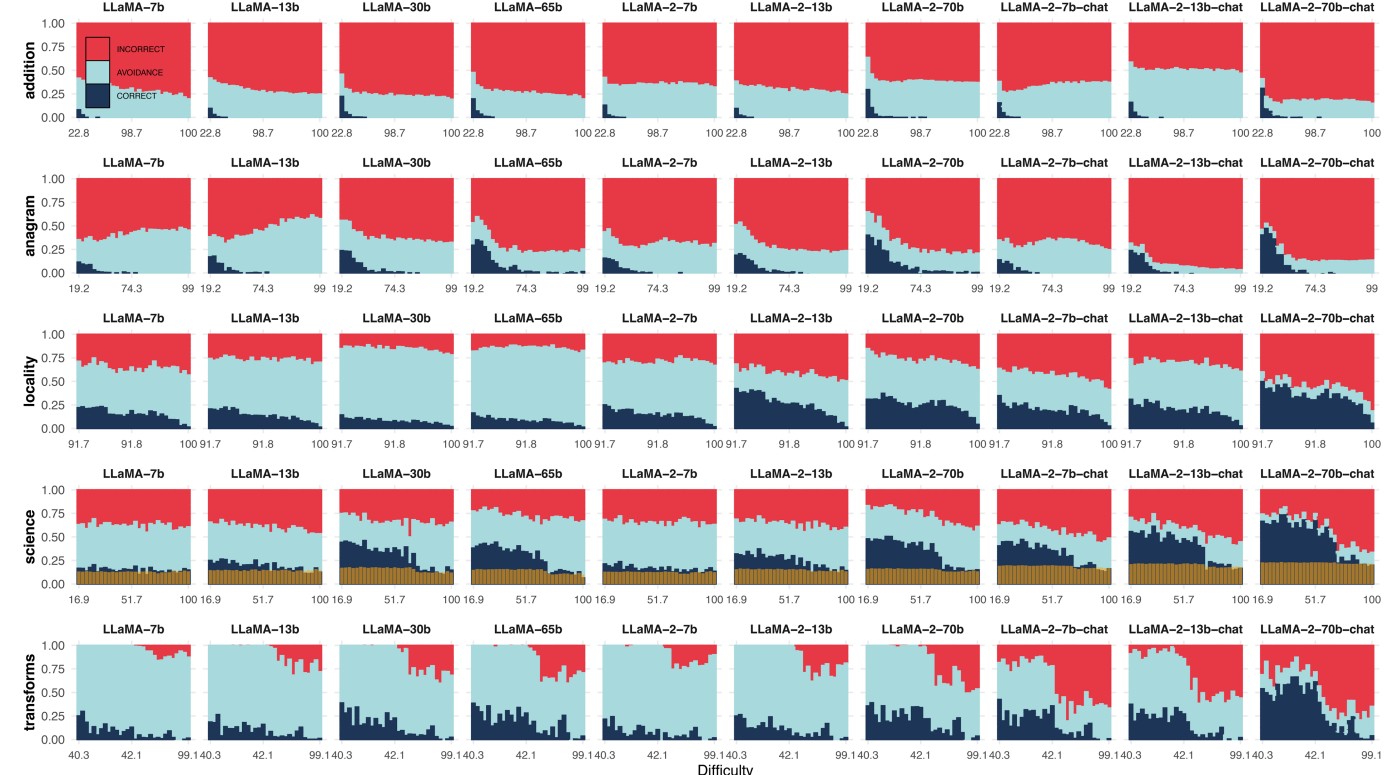

**Extended Data Fig. 2 | Performance of LLaMA models over difficulty.** The values are split by correct, avoidant and incorrect results. Details as in Fig. 2.

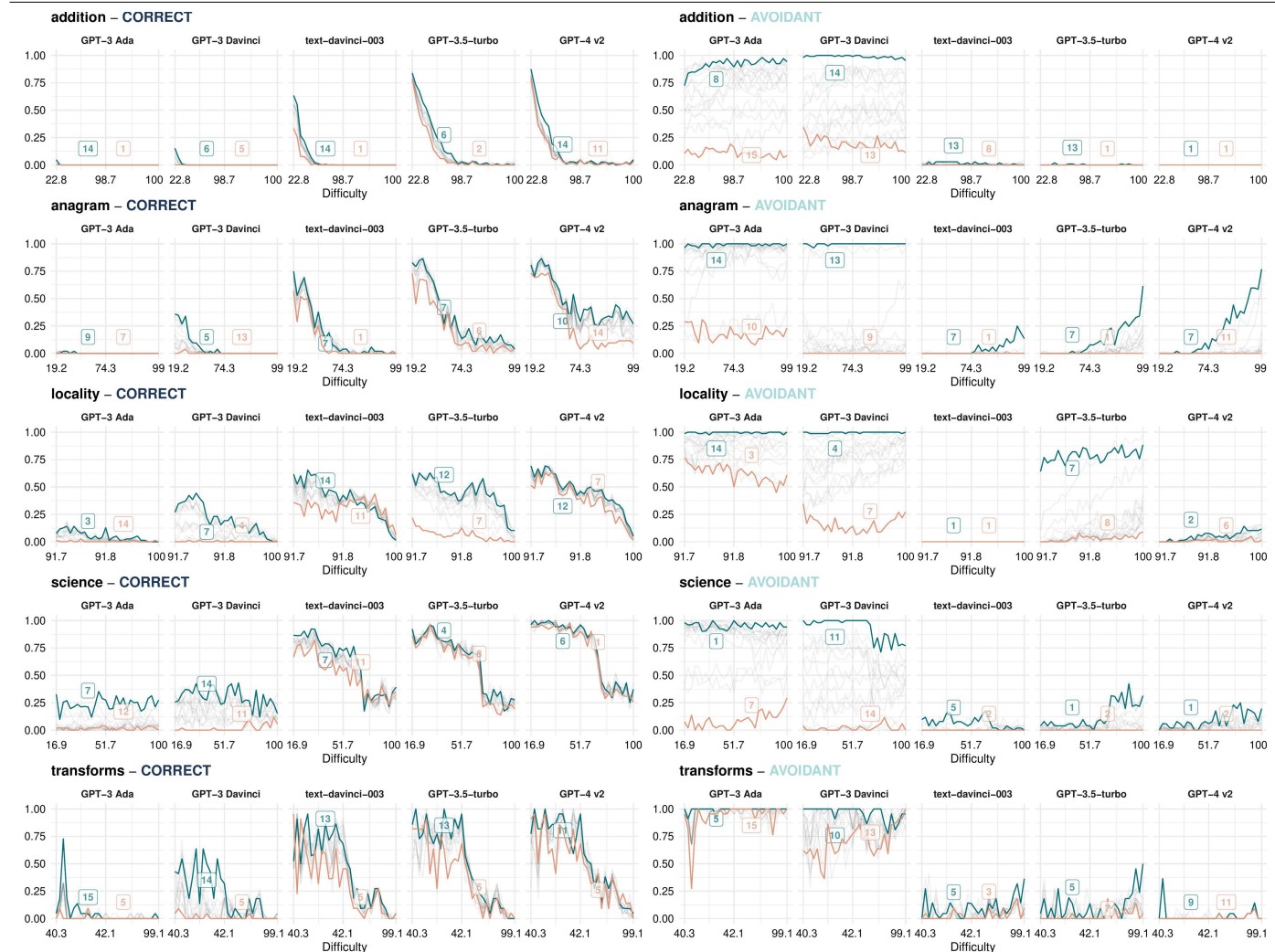

**Extended Data Fig. 3 | Prompting stability of GPT models over difficulty.** Proportion of *correctness* and *avoidance* represented as (grey) curves over difficulty for the 15 prompt templates for the GPT models addressing each of the five benchmarks. The **green** and **bronze** curves correspond to the prompt template that has, respectively, the highest and lowest average correctness, avoidance, or incorrectness. The two small numbers in green and bronze in the plot identify them (corresponding to the template codes in Supplementary Tables 1 and 2). The plots for all the models and all response categories are in section 9 of the Supplementary Information. The same plot for the BLOOM family is in section 11 of the Supplementary Information.

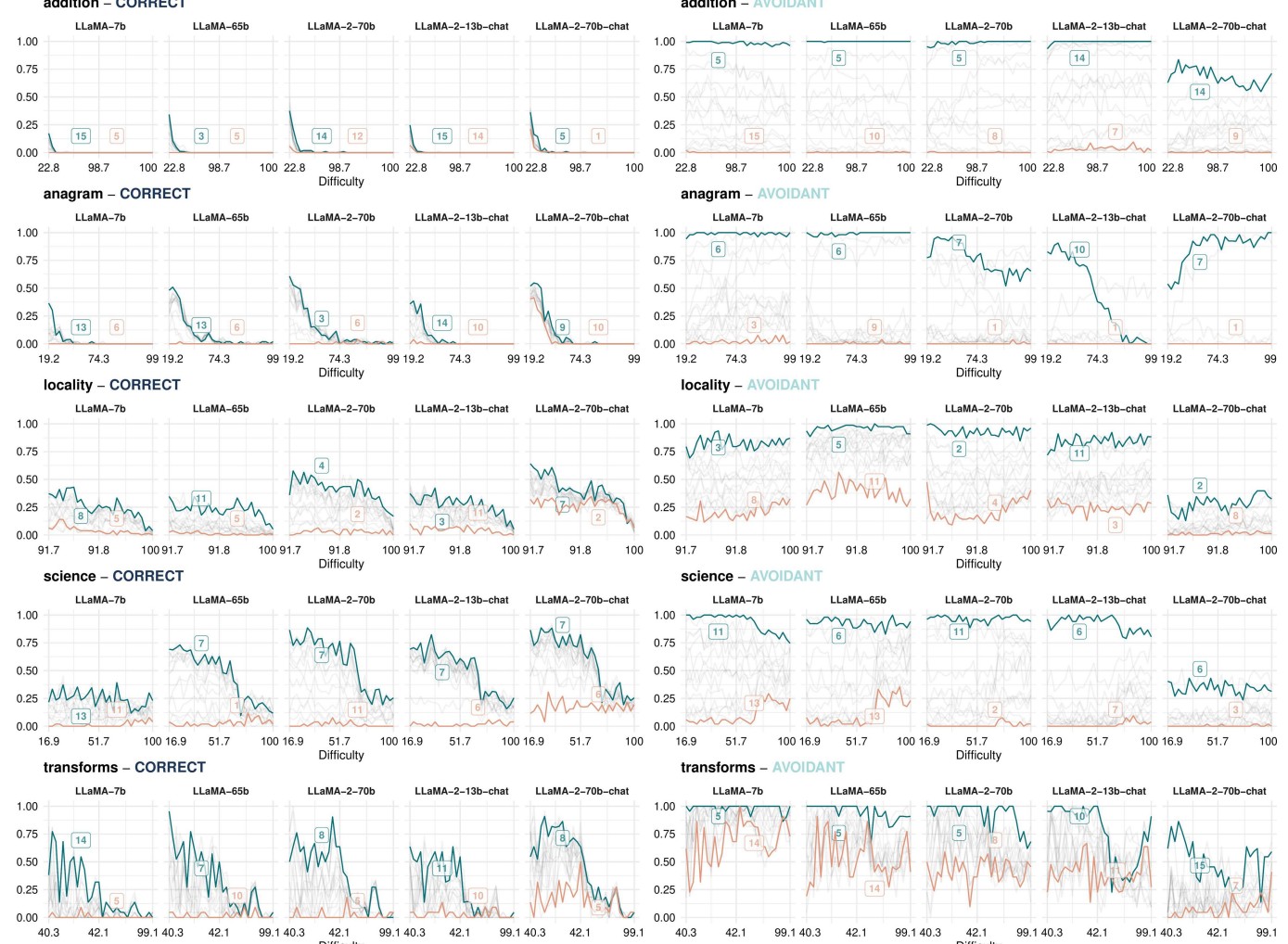

**Extended Data Fig. 4 | Prompting stability of LLaMA models over difficulty.** Proportion of *correctness* and *avoidance* represented as (grey) curves over difficulty for the 15 prompt templates for the LLaMA models addressing each of the five benchmarks. Details as in Extended Data Fig. 3. The plots for all the models and all response categories are in section 9 of the Supplementary Information. The same plot for the BLOOM family is in section 11 of the Supplementary Information.

**Extended Data Table 1 | Proportions, Difficulty Concordance and Prompting Stability for the three families**

| Model | Correctness (c) | | | Prudence (c + a) | | |
|---|---|---|---|---|---|---|
| | Proportion c / (c + a + i) | Difficulty Concordance | Prompting Stability | Proportion (c + a) / (c + a + i) | Difficulty Concordance | Prompting Stability |
| GPT-3 Ada | 2.17 | 46.08 | 27.86 | 85.44 | 9.14 | 9.62 |
| GPT-3 Babbage | 3.41 | 42.40 | 26.08 | 73.82 | 8.19 | 12.81 |
| GPT-3 Curie | 4.96 | 39.67 | 26.09 | 65.14 | 7.74 | 13.38 |
| GPT-3 Davinci | 8.11 | 27.74 | 35.24 | 63.07 | 7.91 | 22.33 |
| text-davinci-001 | 19.83 | 19.87 | 58.34 | 31.34 | 9.98 | 47.78 |
| text-davinci-002 | 28.67 | 14.49 | 65.47 | 34.23 | 9.62 | 56.35 |
| text-davinci-003 | 30.94 | 14.54 | 70.19 | 32.50 | 12.93 | 68.17 |
| GPT-3.5-turbo | 37.15 | 10.92 | 70.71 | 43.50 | 10.11 | 64.42 |
| GPT-4 v1 | 42.08 | 10.23 | 72.86 | 44.63 | 10.30 | 69.67 |
| GPT-4 v2 | 44.39 | 9.98 | 76.80 | 46.54 | 10.23 | 74.28 |
| LLaMA-7b | 7.92 | 29.30 | 46.16 | 58.77 | 7.77 | 31.52 |
| LLaMA-13b | 8.80 | 26.73 | 44.13 | 60.35 | 7.61 | 31.10 |
| LLaMA-30b | 11.33 | 22.03 | 48.75 | 60.26 | 8.02 | 32.64 |
| LLaMA-65b | 12.42 | 20.31 | 35.38 | 59.93 | 8.14 | 21.04 |
| LLaMA-2-7b | 8.16 | 26.18 | 32.31 | 59.18 | 7.53 | 15.84 |
| LLaMA-2-13b | 12.86 | 23.96 | 35.44 | 54.02 | 7.72 | 21.47 |
| LLaMA-2-70b | 17.67 | 17.73 | 38.32 | 59.68 | 7.95 | 24.19 |
| LLaMA-2-7b-chat | 13.96 | 24.27 | 39.85 | 47.83 | 6.73 | 19.26 |
| LLaMA-2-13b-chat | 15.78 | 22.81 | 41.29 | 51.62 | 8.16 | 22.46 |
| LLaMA-2-70b-chat | 25.09 | 18.52 | 57.22 | 38.83 | 7.72 | 32.35 |
| BLOOM-560m | 1.16 | 58.50 | 31.14 | 87.01 | 10.05 | 23.26 |
| BLOOM-1b1 | 1.74 | 58.05 | 30.26 | 85.78 | 9.63 | 22.91 |
| BLOOM-1b7 | 2.66 | 49.90 | 32.14 | 75.08 | 9.20 | 22.17 |
| BLOOM-3b | 2.82 | 48.85 | 25.69 | 81.41 | 9.14 | 14.95 |
| BLOOM-7b1 | 4.20 | 43.20 | 35.86 | 78.99 | 8.26 | 23.89 |
| BLOOM-176b | 7.05 | 33.60 | 39.92 | 69.20 | 7.63 | 26.76 |
| BLOOMz-560m | 6.35 | 44.66 | 55.11 | 40.41 | 7.03 | 34.69 |
| BLOOMz-1b1 | 5.80 | 45.57 | 50.45 | 41.34 | 6.62 | 27.78 |
| BLOOMz-1b7 | 9.09 | 41.55 | 52.70 | 49.98 | 7.07 | 31.06 |
| BLOOMz-3b | 12.06 | 38.84 | 53.38 | 44.49 | 8.25 | 42.04 |
| BLOOMz-7b1 | 13.66 | 36.28 | 56.14 | 46.42 | 7.35 | 39.39 |
| BLOOMz-176b | 17.09 | 30.24 | 60.25 | 46.21 | 7.64 | 49.69 |

Both the correctness and prudence (correctness+avoidance) variants are included. All values in the range 0 to 100, and the higher the better. Visualisation in Fig. 1.