## [Peer Review file · Nature]

Manuscript Title: Larger and More Instructable Language Models Turned Less Reliable

Reviewer Comments & Author Rebuttals

Reviewer Reports on the Initial Version:

Referees' comments:

Referee #1 (Remarks to the Author):

Key results

The paper offers three tasks with controllable difficulty: addition, anagrams, and a restricted geographical query. Several of the key results relate to item difficulty:

- For most task/model combinations, difficulty aligns with accuracy (F1a). However, scaling up tends to increase performance on the more difficult problem instantiations rather than making the easier instantiations error-free (F1b).
- LLMs can refuse to answer questions, but this is generally more dependent on model-specific factors rather than question difficulty (F2).
- Performance depends on superficial details of the prompt (F3).

Validity

The results generally seem valid within the scope of the experimental setup in the paper, but it is not clear how to generalize from these three tasks, which are fairly artificial, to more realistic usage settings. In particular, the failure of LLM pretraining to support strong arithmetic and string-manipulation capabilities is well known (see below), and does not necessarily imply failure on more realistic information-seeking queries.

Originality and significance

The paper is most original in its systematic analysis of some of the most well-known LLMs. The relationship between difficulty, accuracy, and confidence is shown more clearly here than in prior work. However, there is considerable prior work on many parts of the paper:

- On the relationship between difficulty and accuracy, see below for examples. There is also a growing body of prior work on alternative confidence estimation techniques, e.g. <https://arxiv.org/abs/2302.09664>. This paper makes an original contribution by enabling systematic fine-grained control over difficulty, albeit in a restricted setting.
- On answer avoidance, again there is a significant body of work on training LLMs to selectively answer questions, e.g. <https://arxiv.org/abs/2207.05221>. It is more typical to relate answer refusal to accuracy, resulting in precision/recall tradeoff curves.
- On robustness to prompt formulations, there is prior work albeit on smaller instruction-tuned

models, e.g. <https://arxiv.org/abs/2306.11270>

In addition, Chen et al (<https://arxiv.org/abs/2307.09009>), compare the performance of two versions of GPT-3.5 and GPT-4 in March and June 2023, corresponding almost exactly to the GPT-4v1 and GPT-4v2 in this submission. This submission goes further in assessing a broader family of models, and focusing more directly on the relationship between accuracy and coverage (i.e. what fractions of questions the system tries to answer).

Data & methodology

In general, the methodology is solid although as noted above I am not convinced about the generality of the results. One small issue: the submission defines "prompt robustness" as the property that the correctness of the answer is not affected by the prompt. This implies that giving two different incorrect answers is more robust than giving a correct answer and an incorrect answer. In any case, prior work has focused on the related concept of robust accuracy (e.g., <https://arxiv.org/abs/1805.12152>), which could be generalized to recall and precision.

Appropriate use of statistics

I have no concerns

Conclusions

As noted above, I accept the conclusions within the scope of the tasks presented here, but I'm skeptical about the generalization to more typical usage cases.

Improvements

In general, the paper does not engage very much with the prior literature on LLMs, mostly citing this literature through surveys (e.g., Zhao et al 2023; Yang et al 2023; Liu et al 2023). The paper would benefit from engaging more with prior work on LLM calibration and confidence, robustness, and change over time.

References

On LLMs and their weaknesses at mathematical reasoning: <https://arxiv.org/abs/2205.11916> is just one example, which evaluates text-davinci-002 on six mathematical reasoning datasets.

On string manipulation:

<https://arxiv.org/abs/2309.13638>, which is also a useful reference in regard to item difficulty, because it shows how GPT 3.5 and GPT 4 error rates relate to properties of the autoregressive pretraining task.

Some additional citations related to robustness to prompt reformulations:

<https://arxiv.org/abs/2110.08207>

<https://arxiv.org/abs/2204.07705>

On LLM accuracy versus difficulty for humans:

Comparative analysis of large language models in the Royal College of Ophthalmologists fellowship exams (<https://www.nature.com/articles/s41433-023-02563-3>)

Assessing the Accuracy and Reliability of AI-Generated Medical Responses: An Evaluation of the Chat-GPT Model (<https://www.ncbi.nlm.nih.gov/pmc/articles/PMC10002821/>)

Clarity and context

Figure 1 and Table A.7 are quite difficult to understand. Part of the issue is terminology: the top and bottom parts of the figures are labeled "useful/safe" and also "correctness/error-freeness" but actually I think these concepts are better described by well-known existing terminology of recall and precision.

The mathematical notation for correctness is also non-standard and in my opinion unintuitive -- \oplus and \ominus are more typically treated as generalized addition and subtraction operators. The relationship of correctness and error-freeness to "prompt robustness" and "difficulty concordance" was not understandable from the caption; I have concerns about the definition of robustness that I mentioned above.

I didn't understand if the difficulty in the "locality" question is related to the city in the question or in the answer. Both seem relevant.

Referee #2 (Remarks to the Author):

Large language models (LLMs) like ChatGPT have captivated public attention. The ease by which these systems generate natural language, converse with users, and seemingly solve a wide-range of problems has teased the potential that they can radically transform many domains – from the ways students learn to how patients interact with doctors. Yet, how reliable are these systems for such users? In this work, the authors explore the reliability of a suite of LLMs – focusing particularly on the impact of two trends in LLM development: scaling these models, and "shaping" them (via instructions and human feedback). The authors consider three varied model families (GPT, LLaMA, BLOOM), over three tasks (arithmetic, anagram puzzles, and trivia-style locality questions), along three different dimensions of reliability (predictability according to difficulty level, tendency towards avoidance behavior, and robustness to prompt alterations). From their empirical investigations, the authors conclude that shaped up models have unpredictable error patterns, which raises caution for users and developers of these systems alike.

First, I applaud the authors for calling attention to this important problem: updating and scaling these models does not guarantee their reliability. Ensuring predictable reliability is paramount for human users to build good mental models of when they can be appropriately used [1, 2]. I believe the authors' study, which frames and brings this issue to light, will be of interest (or certainly ought to be of interest) to researchers and practitioners across many disciplines – as this new age of LLMs is poised to touch all. I also find the authors' delineation and disentanglement of two trends of LLM development ("scaling up" and "shaping up") clarifying and think it can help contribute positively to the collective language around how we describe these models and their evolution when discussing how such systems ought to be developed, deployed, and regulated. Further, I believe the metrics the authors explore reliability according to – "difficulty concordance", frequency of avoidance, prompt robustness – are in principle valuable dimensions to evaluate. Taken together, the unpredictable landscape of errors that these authors find, and the intriguing large decrease in "avoidant" responses on these rote tasks, are interesting observations and may surprise some readers. I also applaud the others on considering a varied suite of model families; the choice of GPT, LLaMA, and BLOOM are all sensible and valuable to offer deeper investigation into (especially GPT and LLaMA given their popularity, as the authors note).

However, I maintain several concerns about the paper in its current form. My concerns fall into three interrelated buckets: concern about the difficulty proxies, the task selection more broadly, and the interpretation of the results. I address each in turn.

Difficulty measures:

First and most pressingly, I have deep concerns about the validity of the authors' difficulty metrics. Based on my understanding of the authors' description of their methodology in Section 4.1 and Appendix A.4, the authors curate and create a list of possible proxies of human difficulty for their tasks and then only use the difficulty metric which is most correlated with final performance. To quote the authors: "in the end, as all of them are created to be human-like, we chose the most predictive" (line 309-310). I find this method highly problematic. Selecting a metric so that it is predictive, and then claiming that a metric is a valuable predictor (claim F1a in Section 3, line 11, and

line 123, for instance) is circular. A quick look at Table A.6 reveals that many of the difficulty metrics the authors originally considered are actually not particularly correlated with model performance (especially for the BLOOM family). In fact, some difficulty measures have zero (!) correlation with performance. My understanding is that these correlations (in Table A.6) are over the same test instances used in the rest of the paper? As such, I believe selecting the metric which is most predictive and then claiming that the correctness can be predicted by difficulty is effectively cherrypicking. Unfortunately, many of the analyses in the paper are based on sorting the tasks by difficulty, using the effectively-cherrypicked metric per task, which worries me about how much can be drawn about correlations and trends in the other metrics that fall out of the difficulty sorting.

Instead, it may have been more appropriate for the authors to aggregate the correlations across all considered difficulty measures. Additionally, it is not clear to me that the difficulty measures the authors choose for the locality metric actually correlate with real human difficulty (which seems, in part, to be the authors' claims); I'm surprised none of the submetrics the authors consider incorporate the distance value, alongside information about the cities, as I would think that could impact human difficulty? I hesitate then to call this particular metric a "human difficulty metric" (line 96) without cross-verification that it does correlate with humans' perceived difficulty. However, I am not an expert on the actual difficulty measures. My principal concern here is the selection of the metric which is most predictive, and then claiming throughout that correctness tracks (or doesn't track) with difficulty.

Tasks:

In addition to my concerns about the difficulty measures used/selected which are unfortunately quite endemic to the paper given how most analyses build on the difficulty-sorting (see Fig 2, x axis), I also have some concern over the tasks considered. The authors pitch the paper as investigating the "user-driven reliability" (line 48) and usefulness of families of LLMs. The authors justify their tasks as being more realistic to how real users will engage than the multiple-choice questions that often make up LLM benchmarks. Yet, the tasks that the authors consider are quite contrived, in my opinion; with the exception of arithmetic, which is most clearly important to downstream user needs (say, if used in any mathematical setting), it is less clear the importance of the anagram and locality tasks. As such, I again have caution in the generalizability of these findings. How do these reliance measures track across other, real-world domains (especially of the kinds the authors seem motivated by?) And critically, how do they track against real human user beliefs and experiences? The authors note in lines 207-209: "we are urged to analyse how reliable these models are according to the expectations of a regular human user, prompting the system in natural ways," yet no human users are consulted or incorporated in this study?

This is not to say that understanding model behavior on these kinds of tasks, without external human validation, is not valuable; however, for the authors to draw broader conclusions about whether we should change the ways we scale up / shape up these models, I think that more realistic and open-ended tasks of the kinds that humans may actually use these systems for are needed. Ideally, the authors would include at least one more open-ended domain that is more realistic of how users may actually interact with these models (say, asking for help on some educational tasks

that relies on “general world knowledge”). As the authors note, two issues may arise: the difficulty measure may be less clear (but, as mentioned above, I think the current method for selecting such measures is flawed), and the open-ended nature of the responses may be harder to evaluate. I do not think the latter should prohibit the authors from exploring a more ecologically valid task; for instance, a crowdsourcing study could be run to recruit humans to evaluate. While of course this would take time and could potentially get expensive, I do think some aspect of external human user studies is important to sharpen the claims made in the paper about how scaling-up / shaping-up impact “user-driven” reliability. Inclusion of at least one open-ended domain would also permit a more nuanced investigation to the impact of avoidance. I do not believe the current paper has sufficient discussion around the proportion and type of avoidance behavior that may arise in other safety-critical user-facing domains like education, medicine, etc, where we may want it... I would hypothesize shaping-up may increase avoidance behavior frequency, not decrease it... (though this would be good to explore!)

Secondly, focusing just on the tasks that the authors selected — I actually do think that the tasks are confounded (line 252)? In particular, both the Addition and Locality tasks rely on an ability to “understand” and work with numbers. As such, it is possible that failures across tasks can be attributed to issues of these models with numerosity. It would be good for the authors to briefly discuss this.

Interpretation of the results:

Finally, and perhaps easiest to address, I feel that the authors’ interpretation of the results neglects to adequately disentangle performance along the dimensions the authors nicely laid out in their Introduction: not just that these models are being shaped up, but that they are being scaled up. The authors focus primarily on the impact of shaping up these models; understandably, this seems to have the biggest impact on the performance metrics considered. However, what about the impact of scaling? For instance, if you look directly at the LLaMA instances and compare models with or without shaping up, but different parameter sizes (e.g., LLaMA-2-7b-chat vs LLaMA-2-13b-chat vs. LLaMA-2-70b-chat). Such investigations can help researchers gain deeper insight into the impact of scaling, and I felt not adequately digging into such results was a lost opportunity in this first paper version. Again, though, I feel that before deeper analyses can be done along those lines, the issue of the difficulty measures ought to be addressed.

Overall, given these limitations, I think that it is not appropriate to draw – from these current results – that we need a change in the direction of scaling up and shaping up these models (lines 249-250). I think adequately addressing such claims needs at least one more open-ended task, and ideally, the inclusion of real humans “in-the-loop” of evaluating the output. Again, I believe that the high-level problem statement and research questions that the authors lay out are of value and interest to many research communities. I also applaud the authors on their granular de-aggregated analyses. I certainly found the piece interesting and believe that, if executed rigorously with appropriate distance metrics and ideally an additional ecologically valid task, the work could bring value to many communities. I also reiterate that it is no easy feat to evaluate this many models, over this many problems, and admire the authors’ courage to take on that task.

Finally, I leave a few specific and minor comments/questions that I hope will improve a future iteration of the authors' work:

- I am confused as to what this "human expectations" dataset would encompass (Section 3).

Expectations of what? Model behavior?

- I appreciate the authors' avoidance taxonomy and found it helpful. However, I want to raise attention that the example given for C1 does include a first-person singular pronoun? I found the definition therein confusing and perhaps contradictory. Additionally, I found the delineation of the impact of certain forms of avoidance arising from "shaping up" as also perhaps a bit imprecise; shaping up may change avoidance behavior in ways beyond getting the model to say "I am not a language model." Shaping up could also, in other domains – say that are more open-ended and ecologically valid as I touched on above – could lead to more of the kind of avoidance without first person "I". I'd encourage a renaming of that part of the categories.

- When was GPT-3.5-turbo accessed? It would be good to add the date(s). I believe that this model may be continually updated by OpenAI. The updates can matter [3]. I appreciate that the authors listed the exact versions of GPT-4 used. Additionally, and a minor note: it would be good for the authors, for reproducibility, to note in the Supplement what mode of interaction they used with the various APIs, particularly GPT. For instance, did the authors use the Completion or ChatCompletion APIs? Was a "system-level" prompt provided; if so, what was it?

- The authors source prompt templates from pre-existing textbook / web links. As such, it's possible – I'd say highly probable – that these text snippets from which prompts are derived have been included in the web-scraped training data of these models. Are there trends the authors find in the kinds of prompts that models are least robust to? Such an analysis could be nice, but is not necessary to the overall story.

- I am confused on the authors' conclusions about what should be done to improve reliability. I hesitate to draw the conclusion that training "should" give more relevance to easy problems; it's not immediately clear that this will improve the (lines 224-225). And lines 228-230 call for just more shaping up (fine-tuning)?

- Lastly, the authors discuss three kinds of "scaling up" in the introduction – number of parameters, amount of training data, and training time. To my reading, the authors do not discuss the impact of training time at all? Nor do the authors break down the impact of scaling along number of parameters vs. amount of training data. This is fine, but it could be worth noting that the different forms of scaling up could be conflated and may have differential impact on reliability that this study is not able to address (but are ripe grounds for future work).

[1] Bansal, G., Nushi, B., Kamar, E., Lasecki, W. S., Weld, D. S., & Horvitz, E. (2019). Beyond Accuracy: The Role of Mental Models in Human-AI Team Performance. Proceedings of the AAAI Conference on Human Computation and Crowdsourcing, 7(1), 2-11.

[2] Bansal, G., Nushi, B., Kamar, E., Horvitz, E., & Weld, D.S. (2021). Is the Most Accurate AI the Best Teammate? Optimizing AI for Teamwork. AAAI Conference on Artificial Intelligence.

[3] Chen, L., Zaharia, M., & Zou, J. (2023). How is ChatGPT's behavior changing over time?. arXiv preprint arXiv:2307.09009.

Scaled-up, Shaped-up, but Letting Down? Reliability Fluctuations of Large Language Model Families

Revision Letter - April 2024

We thank the editor and reviewers for their feedback. In this letter, we provide a detailed account of how we have addressed their comments. We first summarise the most relevant changes:

- Increased generalisability of results:** We have now added two new benchmarks covering a wide range of tasks in several real-world domains and different degrees of open-endedness, while keeping several requirements: a sufficient number of instances for the robustness of analysis, a ground-truth format that makes difficulty meaningful, and no confounding factors to difficulty coming from possible user hints. To cover ecological diversity and validity, new prompt variations were created based on guidelines from reviewer R1 and recent literature on realistic prompt usage. This broadens the scope of the experiments and strengthens the generalisability of our findings.
- Human-validated choice of difficulty metrics:** We conducted a human study (S1) on both perceived and actual (human) difficulty across all tasks in order to refine our choice of difficulty metric. The study involved more than 2800 responses from a total of $N=189$ subjects, from whom we capture the perceived expected difficulty, their actual performance, and the confidence in their solution. In response to reviewer R2's criticism, we now select the difficulty metrics that best match human expectations per task, to better underpin our analysis of reliability. The consistent correlations between these difficulty proxies and model results prove the robustness of our methodology.
- Evidence of unreliability and overreliance when humans evaluate models.** We performed a study S2 with 10800 responses from $N=300$ subjects, where the participants had to label model responses as correct, incorrect or avoidant; participants could also express uncertainty. This follows reviewer R2's suggestion of exploring ecologically-valid scenarios in which humans assess model outputs, and may rely on them or not. The results confirm the poor quality of non-expert human scoring and the significant risk of overreliance. There is virtually no region of difficulty without incorrect responses mistakenly seen as correct by humans, potentially being misused. This is further motivation and justification for this work.
- Improved contextualisation of the contributions and impact:** The revised manuscript now includes a more detailed analysis of the literature on the effect of task difficulty in language models, avoidance (refusal) strategies, and the robustness of language models to prompt variations, as suggested by R1. As opposed to the related literature, our paper takes a distinctive and comprehensive perspective on the evolution on LLMs reliability: a) analyses scaling and shaping (now separately, as indicated by R2), b) introduces intrinsic instance-level difficulty metrics (now validating their alignment with humans expectations), and c) explores prompt robustness in terms of difficulty and for three-valued labels.

Despite the significant increase in new material, the paper has been streamlined with an appropriate balance of details and references between the main paper and the appendix, keeping a similar size as the previous version. We are of course open to further rearrangement of figures and tables.

As for the time pressure of a fast-changing environment in LLMs science and technology, the acquired perspective since the first submission has strengthened the extra scientific value of keeping a more abstract analysis of the evolution of language families (including the now-discontinued early GPT3 models, which we managed to run before the cutoff on January 4th). This has made our methodology and findings even more timely now than when submitted one year ago. We are eager to share the extensive repository of benchmarks, results and tools emanating from our project for other initiatives tracking the reliability of future LLMs.

The rest of this letter includes a point-by-point response to the reviewer's comments.

The authors.

Editor

Your manuscript, "Scaled-up, Shaped-up, but Letting Down? Reliability Fluctuations of Large Language Model Families", has now been seen by 2 referees, whose comments are attached below. Thanks for your patience as we gathered these reports - we were waiting on a 3rd who has unfortunately not provided one, but we believe that these two are relatively consistent and enough for us to proceed with our decision for now.

While they both find the work of clear interest, as indeed do we, they have raised substantial concerns that first need to be addressed before we can consider the paper further for possible publication in Nature.

As you will see from the comments, there are essentially two overarching issues raised by the reviewers.

E.1. First, raised by both reviewers, the range of tasks here are very useful in their enablement of a systematic approach, but there are substantial concerns about the generalisability of the observation to more 'realistic' tasks which typical users might perform. We agree with R2 that "Ideally, the authors would include at least one more open-ended domain that is more realistic of how users may actually interact with these models".

Answer. We have now included many new tasks to better support the ecological validity of our analysis, covering a range of domains (education, science, general-world knowledge, etc.) with questions that are diverse in their outputs (from a few characters to hundreds of words) but objective enough to have a ground truth to allow us to evaluate the quality of human assessment and verification (as explored by the human study S2). We have arranged them into two new benchmarks for the sake of clarity of exposition but also because we now illustrate the methodology, including the difficulty metrics, with *benchmarks*, now composed of multiple tasks. The first benchmark is CommonTransforms, simply referred to as **transforms**, which integrates existing and new tasks on a diversity of domains that cover a wide range of transformation, formatting and wrangling operations with text that constitute a very common use of LLMs. The tasks go from simple transformation effort involving a few characters to more structural or semantic transformations involving extensive rewriting of long text, with size and depth of transformation being proxies for difficulty (validated by human study S1). The second benchmark is Gradual Science, simply referred to as **science**, and is built by integrating questions from OpenBookQA, which includes elementary-level science questions, and GPQA, which poses more challenging graduate-level questions. This also ensures a wide range of difficulty, which in this case is based on a human proxy (also validated and calibrated by S1).

The **transforms** benchmark is as open-ended as possible, taking into account that we need an objective ground truth to evaluate the quality of human verification, compared to more creative tasks, such as writing resumes, summaries, poems, etc., where the grading must subjectively come from humans —more on this later. The outputs for **transforms** may require extensive elaboration of the input (hundreds of characters) to form a correct answer, which can also be hundreds of characters long. The aim was to simulate as closely as possible the complexity and depth of real-world questions in a controlled experimental setting.

The selection of this benchmark together with more conventional, multiple-choice questions such as those included in the **science** benchmark is strategic, designed to test the reliability of AI models across a spectrum of tasks of different degrees of open-endedness. It is important to note that in all cases, including multiple-choice questions, the output from the LLMs is open-ended, not constrained in any way, and may include long responses with chain-of-thought derivations or explanations. For all tasks, this has to be parsed into correct, incorrect or avoidant scores.

In general, **all these tasks were chosen to reflect the typical real-world use of LLMs¹**, in data wrangling scenarios (involving dates, phone numbers, names, emails), as investigated in numerous studies on the capabilities of LLMs in data transformation tasks^{2,3,4,5,6,7,8,9,10,11,12,13,14,15,16}, other realistic tasks from various domains (see Figure 3 in Zheng et al.¹ for a study of typical uses of LLMs) such as education, world knowledge, information retrieval, advertising, administration, coding, scheduling and shopping, as well as science questions of varying complexity. This, together with the old benchmarks, gives a more representative view of the tasks users do with LLMs.

The new experiments total 1544 questions for **science** and 730 instances for **transforms**. These, jointly with the old ones, and multiplied by the number of prompts (15) and models (32) makes more than 4 million responses. This amount is necessary for the robustness and generalisability of results, but at the same time emphasises why we need questions with ground truth for which algorithmic grading is possible. See our response to R.1. for further details.

The extended plots and metrics with the new benchmarks have corroborated the findings in the previous version. The only remarkably different result that happens with the two new benchmarks is that we find an area of low difficulty for **science** with GPT-4 (see Figure 2) with almost perfect results up to medium difficulties. The multiple-choice character of this benchmark and the high possibility of contamination may suggest that this is actually an exception that may have been dominating the general discourse about reliability, instead of all the other cases. Our paper can help realise this.

¹ Zheng, L., Chiang, W. L., Sheng, Y., Li, T., Zhuang, S., Wu, Z., ... & Zhang, H. (2023). Lmsys-chat-1m: A large-scale real-world LLM conversation dataset. arXiv preprint arXiv:2309.11998.

² Narayan, A., Chami, I., Orr, L., Arora, S., & Ré, C. (2022). Can foundation models wrangle your data?. arXiv preprint arXiv:2205.09911.

³ Zhang, H., Dong, Y., Xiao, C., & Oyamada, M. (2023). Large language models as data preprocessors. arXiv preprint arXiv:2308.16361.

⁴ Suhara, Y., Li, J., Li, Y., Zhang, D., et al. (2022). Annotating columns with pre-trained language models. In Proceedings of the 2022 International Conference on Management of Data (pp. 1493–1503).

⁵ Deng, X., Sun, H., Lees, A., Wu, Y., et al. (2020). TURL: Table understanding through representation learning. Proceedings of the VLDB Endowment, 14(2020), 307–319.

⁶ Iida, H., Thai, D., Manjunatha, V., Iyyer, M. (2021). TABBIE: Pretrained representations of tabular data. In Proceedings of the 2021 Conference of the North American Chapter of the Association for Computational Linguistics: Human Language Technologies (pp. 3446–3456).

⁷ Jaimovitch-López, G., Ferri, C., Hernández-Orallo, J., Martínez-Plumed, F., et al. (2023). Can language models automate data wrangling? Machine Learning Journal, Volume 112, pages 2053–2082.

⁸ Badaro, G., Papotti, P. (2022). Transformers for tabular data representation: a tutorial on models and applications. Proceedings of the VLDB Endowment, 15, 3746–3749.

⁹ Jiao, X., Yin, Y., Shang, L., Jiang, X., et al. (2020). TinyBERT: Distilling BERT for natural language understanding. In Findings of the Association for Computational Linguistics: EMNLP 2020 (pp. 4163–4174).

¹⁰ Tang, J., Zuo, Y., Cao, L., Madden, S. (2022). Generic entity resolution models. In Proceedings of the Advances in Neural Information Processing Systems 2022 First Table Representation Workshop.

¹¹ Peeters, R., Bizer, C. (2023). Using chatgpt for entity matching. In Proceedings of Advances in Databases and Information Systems.

¹² Korini, K., & Bizer, C. (2023). Column Type Annotation using ChatGPT. arXiv preprint arXiv:2306.00745.

¹³ Li, P., He, Y., Yashar, D., Cui, W., Ge, S., Zhang, H., ... (2023). Table-GPT: Table-tuned GPT for diverse table tasks. arXiv preprint arXiv:2310.09263. Retrieved from <https://arxiv.org>

¹⁴ Cong, T., Hulsebos, M., Sun, Z., Groth, P., ... (2023). Observatory: Characterizing embeddings of relational tables. arXiv preprint arXiv:2310.07736. Retrieved from <https://arxiv.org>

¹⁵ Allen, B. P., Stork, L., Groth, P. (2023). Knowledge engineering using large language models. arXiv preprint arXiv:2310.00637. Hoseini, S., Theissen-Lipp, J., Quix, C. (2023). Semantic data management in data lakes. arXiv preprint arXiv:2310.15373. Zhang, H., Dong, Y., Xiao, C., Oyamada, M. (2023). Jellyfish: A large language model for data preprocessing. arXiv preprint arXiv:2312.01678.

¹⁶ Feuer, B., Liu, Y., Hegde, C., Freire, J. (2023). ArcheType: A novel framework for open-source column type annotation using large language models. arXiv preprint arXiv:2310.18208. Heidari, A., McGrath, J., Ilyas, I. F., & Rekatsinas, T. (2019, June). Holodetect: Few-shot learning for error detection. In Proceedings of the 2019 International Conference on Management of Data (pp. 829-846).

E.2. And second there are substantive concerns about the suitability of the difficulty metrics and follow-on analysis which should be convincingly addressed. We hope that you are able to rise to the various challenges posed!

Answer. We have undertaken two comprehensive human studies to choose the difficulty metrics and better understand their effect on use reliability. The studies, S1 and S2, conducted through www.prolific.com with the Concerto Platform and authorised by the ethical committee of one of our institutions, had two main goals: 1) estimating human task difficulty (perceived and real) and 2) determining how humans assess language model outputs (also in terms of difficulty). S1 focuses on task inputs (from humans) and S2 focuses on task outputs (from the model).

In S1, for each of the $M=540$ instances, we asked 5 participants the following: first, the perceived difficulty of the instance (Q1) (1 - “probability that an average human gets a correct answer”), then we asked them to solve it (Q2) and then to rate their confidence (Q3). We asked the same Q1 again at the end (Q4). Hence, questions Q1 and Q4 represent the expected difficulty before and after subjects attempt the question. At the same time, Q2 gives us actual difficulty (for this population), whereas Q3 serves as control over Q2 and an indication of self-confidence. As argued in the first version of this paper, deviations of the results of language models from these expectations are what humans would find strange and would severely affect the reliability of these systems. Using these expected difficulties, calculated from Q1 and Q4 in this sample of questions, **we can now choose, for each benchmark, the difficulty metric that correlates the most with this expected difficulty**. This way, in the updated version of the paper, we are ordering the x-axis by a difficulty metric that is a proxy of the difficulty expectations from humans. Also, by comparing the different questions against each other, we get insights about the relation between the expected performance -- both personal and generic -- and actually realised performance.

The second study (S2) simulates a scenario where humans are shown a question ($M=2700$) and the answer produced by a model (simply referred to as “assistant”). The human subject needs to determine if the “assistant” is either correct, incorrect or avoidant. In ecologically-valid scenarios, the user may or may not know the answer, using the language model for automation or for actually solving the question, but there is a range of possibilities in between. The case where the human says the answer is correct but it is actually incorrect is more severe than all the other eight combinations in the confusion matrix. Either when the user interacts with the model in a semi-automated scenario or supervises a sample of model-answered items in a fully-automated scenario, analysing human assessment in terms of kind of error and difficulty contributes to understanding reliability. To this end, instances were selected with a sampling mechanism that ensures a diverse range of difficulty values and an approximately balanced distribution of correctness, incorrectness and avoidance. Looking at the confusion matrices in new Figures A.15 to A.18, we see that grading by non-expert humans is of poor quality, as expected, but now confirmed at the request of R2. More importantly, there is a significant number of incorrect answers being considered correct by humans, the really dangerous situation about reliability, even when humans are supervising or verifying an assistant. Finally, a crucial question is whether the *proportion* of the incorrect-to-correct type of errors (over all incorrect answers) is higher as difficulty increases (new Fig. A.19). If the user expectations on difficulty were aligned with model results, we should have fewer cases on the left area of the curve (easy instances), and those should be better verified by humans, having a safe haven or operating area on those instances that are regarded as easy by humans. But, unfortunately, this only happens for the addition benchmark for very easy instances, and for the anagram benchmark for a wider range, since verification is straightforward in this case.

In sum, the human studies have substantiated the difficulty proxies we are using (highest correlations) and have confirmed the poor quality of non-expert human annotations and the alarming frequency of the incorrect-to-correct type of error (emphasising the reality of the reliability problem).

E.3. I should note that the prior work raised (mostly by R1) should of course be discussed but has not played a key role in our current decision.

Answer. We now include a detailed analysis of prior work, including the references^{17,18,19,20,21,22,23,24} suggested by R1 and more (the detailed coverage in new appendix A.14 but most references are included in the main paper). **The abundant unique elements of our work are now better contextualised** with respect to the study of semantic entropy¹⁷, confidence estimation¹⁸, prompt sensitivity^{19,20,21}, and many other papers analysing the progress and limitations of LLMs^{22,23,24}. See R1.2 below for an extended discussion.

In view of the additional data and analysis that are required to address these concerns, we appreciate that the necessary revisions will probably take some time. But let me assure you that we will nevertheless be happy to look at a revised manuscript (unless, of course, something similar has by then been accepted at Nature or appeared elsewhere).

Please also note that we are committed to providing a fair and constructive peer-review process: do not hesitate to contact us if there are specific requests from the reviewers that you believe are technically impossible or unlikely to yield a meaningful outcome. I should stress, however, that we would be reluctant to trouble our referees again unless we thought their comments and any editorial issues had been responded to in full.

¹⁷ Kuhn, L., Gal, Y., & Farquhar, S. (2023). Semantic Uncertainty: Linguistic Invariances for Uncertainty Estimation in Natural Language Generation. arXiv preprint arXiv:2302.09664.

¹⁸ Kadavath, S., Conerly, T., Askell, A., Henighan, T., Drain, D., Perez, E., ... & Kaplan, J. (2022). Language Models (Mostly) Know What They Know. arXiv preprint arXiv:2207.05221.

¹⁹ Sun, J., Shaib, C., & Wallace, B. C. (2023). Evaluating the Zero-shot Robustness of Instruction-tuned Language Models. arXiv preprint arXiv:2306.11270.

²⁰ Sanh, V., Webson, A., Raffel, C., Bach, S. H., Sutawika, L., Alyafeai, Z., ... & Rush, A. M. (2021). Multitask Prompted Training Enables Zero-shot Task Generalization. arXiv preprint arXiv:2110.08207.

²¹ Wang, Y., Mishra, S., Alipoormolabashi, P., Kordi, Y., Mirzaei, A., Arunkumar, A., ... & Khashabi, D. (2022). Super-NATURALINSTRUCTIONS: Generalization via Declarative Instructions on 1600+ NLP Tasks. arXiv preprint arXiv:2204.07705.

²² Chen, L., Zaharia, M., & Zou, J. (2023). How is ChatGPT's Behavior Changing Over Time?. arXiv preprint arXiv:2307.09009.

²³ Kojima, T., Gu, S. S., Reid, M., Matsuo, Y., & Iwasawa, Y. (2022). Large Language Models Are Zero-shot Reasoners. *Advances in Neural Information Processing Systems*, 35, 22199-22213.

²⁴ McCoy, R. T., Yao, S., Friedman, D., Hardy, M., & Griffiths, T. L. (2023). Embers of Autoregression: Understanding Large Language Models through the Problem They Are Trained to Solve. arXiv preprint arXiv:2309.13638.

Reviewer 1

Key results

The paper offers three tasks with controllable difficulty: addition, anagrams, and a restricted geographical query. Several of the key results relate to item difficulty:

- For most task/model combinations, difficulty aligns with accuracy (F1a). However, scaling up tends to increase performance on the more difficult problem instantiations rather than making the easier instantiations error-free (F1b).
- LLMs can refuse to answer questions, but this is generally more dependent on model-specific factors rather than question difficulty (F2).
- Performance depends on superficial details of the prompt (F3).

Validity

R1.1. The results generally seem valid within the scope of the experimental setup in the paper, but it is not clear how to generalize from these three tasks, which are fairly artificial, to more realistic usage settings. In particular, the failure of LLM pretraining to support strong arithmetic and string-manipulation capabilities is well known (see below), and does not necessarily imply failure on more realistic information-seeking queries.

Answer. The original three tasks are akin to simplified tasks that are used in the evaluation of other artificial and natural cognitive systems. They allow us to isolate the factors we want to analyse from confounders that would be present in other less well-defined tasks. They are very informative for this purpose. But we agree with the reviewer that a wider range of tasks would give extra support and generalisability to our findings. Accordingly, we now include tasks that require a combination of understanding of several parts of the input, seeking, retrieving and synthesising information from multiple sources, and generating responses that go beyond operations, combinatorics or constrained retrieval. In particular, we now include more than seventy new tasks that we group into the following benchmarks, better representing realistic user interactions with LLMs:

- **Common Transforms**, referred to as **transforms** (73 tasks with 10 instances each): Here, we integrate a wide range of information-centric transformation tasks directly rooted in ecologically-valid scenarios. According to Zheng et al.²⁵, the kinds of tasks and domains that are most prevalent in the use of LLMs today are coding, information retrieval, language translation, text generation and transformation, data formatting, business strategy development, product exploration, commerce activities, scheduling and global knowledge acquisition. Following this list, we integrate many questions from Jaimovitch et al.²⁶, a data wrangling dataset, which covers data formatting tasks, but also add many new tasks about world knowledge, information retrieval, advertising, administration, coding²⁷, scheduling and retailing, reflecting a very wide range of real user interactions with language models. This approach ensures that this benchmark captures the everyday use of language models, navigating complex, multifaceted tasks that users commonly undertake.

²⁵ Zheng, L., Chiang, W. L., Sheng, Y., Li, T., Zhuang, S., Wu, Z., ... & Zhang, H. (2023). Lmsys-chat-1m: A large-scale real-world LLM conversation dataset. arXiv preprint arXiv:2309.11998.

²⁶ Jaimovitch-López, G., Ferri, C., Hernández-Orallo, J., Martínez-Plumed, F., & Ramírez-Quintana, M. J. (2023). Can language models automate data wrangling?. *Machine Learning*, 112(6), 2053-2082.

²⁷ Rand Fishkin. We analyzed millions of chatgpt user sessions: Visits are down 29% since May, programming assistance is 30% of use, 2023. URL <https://sparktoro.com/blog>. Accessed: 12-02-2024.

- Gradual Science, referred to as **science** (1,544 instances): In order to complement the original benchmarks and the previous one with more knowledge-intensive ones, and a very popular presentation in educational settings (Q&A), we integrate OpenBook²⁸ a collection of multiple-choice questions in elementary science, based on 1,329 established facts. This database is particularly relevant to the practical application of LLMs in educational settings^{29,30,31,32,33,34}. It includes questions of varying levels of difficulty, as determined by human judgement, allowing for a detailed study of how LLMs match human expectations in responding to domain-specific questions. We sampled 1,000 questions randomly from OpenBook. To complement the benchmark with more advanced science questions, some of them requiring several minutes to solve, we included GPQA³⁵, a dataset containing more than 500 graduate-level questions written by domain experts that challenge LLMs to demonstrate a deep understanding of biology, physics, and chemistry. It mimics the rigorous questions LLMs are likely to encounter in academic and research settings^{36,37,38,39,40,41}, and provides a lens through which to examine their handling of complex, data-rich tasks. The included questions are designed to test the limits of both human experts and state-of-the-art AI systems³⁵ and are Google-proof (simple Internet lookup). In both OpenBook and GPQA, human ratings of difficulty are incorporated, enriching the understanding of model performance relative to human standards.

They are described in the introduction, fully in sections 4.1 and 4.2, and summarised in Table 2, with examples of instances and their raw and human-calibrated difficulties. The 73 tasks in **transforms** are listed in Table A.13. We have applied the same methodology to these new tasks and the results for these benchmarks complement the key figures in the paper or are covered in new figures. These results totally support our previous findings and make them more generally applicable.

Originality and significance

R1.2. The paper is most original in its systematic analysis of some of the most well-known LLMs. The relationship between difficulty, accuracy, and confidence is shown more clearly here than in prior work. However, there is considerable prior work on many parts of the paper:

- On the relationship between difficulty and accuracy, see below for examples. There is also a growing body of prior work on alternative confidence estimation techniques, e.g.

²⁸ Mihaylov, T., Clark, P., Khot, T., & Sabharwal, A. (2018). Can a suit of armor conduct electricity? a new dataset for open book question answering. arXiv preprint arXiv:1809.02789.

²⁹ Newton, P. M., & Xiromeriti, M. (2023). ChatGPT performance on MCQ exams in higher education. A pragmatic scoping review.

³⁰ Meo, S. A., Al-Masri, A. A., Alotaibi, M., Meo, M. Z. S., & Meo, M. O. S. (2023, July). ChatGPT knowledge evaluation in basic and clinical medical sciences: multiple choice question examination-based performance. In *Healthcare* (Vol. 11, No. 14, p. 2046).

³¹ Gonsalves, C. (2023). On ChatGPT: what promise remains for multiple choice assessment?. *Journal of Learning Development in Higher Education*, (27).

³² Xuan-Quy, D., Ngoc-Bich, L., Bac-Bien, N., & Xuan-Dung, P. (2023). LLMs' Capabilities at the High School Level in Chemistry: Cases of ChatGPT and Microsoft Bing Chat.

³³ Li, P. H., Lee, H. Y., Cheng, Y. P., Starčić, A. I., & Huang, Y. M. (2023, August). Solving the self-regulated learning problem: Exploring the performance of ChatGPT in Mathematics. In *International Conference on Innovative Technologies and Learning* (pp. 77-86). Cham: Springer Nature Switzerland.

³⁴ Ngo, A., Gupta, S., Perrine, O., Reddy, R., Ershadi, S., & Remick, D. (2024). ChatGPT 3.5 fails to write appropriate multiple choice practice exam questions. *Academic Pathology*, 11(1), 100099.

³⁵ Rein, D., Hou, B. L., Stickland, A. C., Petty, J., Pang, R. Y., Dirani, J., ... & Bowman, S. R. (2023). Gpqa: A graduate-level google-proof q&a benchmark. arXiv preprint arXiv:2311.12022.

³⁶ Beam, A. L., Drazen, J. M., Kohane, I. S., Leong, T. Y., Manrai, A. K., & Rubin, E. J. (2023). Artificial intelligence in medicine. *New England Journal of Medicine*, 388(13), 1220-1221.

³⁷ Marr Bernard. *Medicine And Wellness* [online] Forbes; 2023. [Accessed April 2023]. Revolutionizing Healthcare: The Top 14 Uses Of ChatGPT.

³⁸ Kung, T. H., Cheatham, M., Medenilla, A., Sillos, C., De Leon, L., Elepaño, C., ... & Tseng, V. (2023). Performance of ChatGPT on USMLE: Potential for AI-assisted medical education using large language models. *PLoS digital health*, 2(2), e0000198.

³⁹ Bryant, S. (2023). Assessing GPT-4's Role as a Co-Collaborator in Scientific Research: A Case Study Analyzing Einstein's Special Theory of Relativity.

⁴⁰ Boiko, D. A., MacKnight, R., Kline, B., & Gomes, G. (2023). Autonomous chemical research with large language models. *Nature*, 624(7992), 570-578.

⁴¹ de Kok, T. (2023). Generative LLMs and Textual Analysis in Accounting:(Chat) GPT as Research Assistant?. Available at SSRN.

<https://arxiv.org/abs/2302.09664>. This paper makes an original contribution by enabling systematic fine-grained control over difficulty, albeit in a restricted setting.

- On answer avoidance, again there is a significant body of work on training LLMs to selectively answer questions, e.g. <https://arxiv.org/abs/2207.05221>. It is more typical to relate answer refusal to accuracy, resulting in precision/recall tradeoff curves.
- On robustness to prompt formulations, there is prior work albeit on smaller instruction-tuned models, e.g. <https://arxiv.org/abs/2306.11270>

In addition, Chen et al (<https://arxiv.org/abs/2307.09009>), compare the performance of two versions of GPT-3.5 and GPT-4 in March and June 2023, corresponding almost exactly to the GPT-4v1 and GPT-4v2 in this submission. This submission goes further in assessing a broader family of models, and focusing more directly on the relationship between accuracy and coverage (i.e. what fractions of questions the system tries to answer).

Answer. Thank you for all these suggestions. We have included all these references in the paper. In addition, in the new section A.14 in the appendix, we have further discussed how our approach aligns with and differs from the state of the art, using these and many other recent papers on these topics. Here, we also discuss those references suggested in R1.6:

On the **relationship between difficulty and performance**, our research extends the concept of difficulty to very different domains, also providing a more granular analysis of model performance across a spectrum of difficulty levels. In our case, difficulty is instrumental for understanding user expectations and seeing the distribution of errors (and avoidance) as a function of difficulty. On a smaller scale, specifically for medical queries, Raimondi et al.⁴² and Johnson et al.⁴³ also study the nuanced relationship between the accuracy of LLMs, such as ChatGPT and Bing Chat, and question difficulty. These two studies above show models perform better on simpler questions, with a notable drop in accuracy as questions become more complex. But none of these studies includes an analysis of the evolution of language model families in how this distribution is changing as models are scaled up and shaped up, which is the main topic of our paper, and with the specific lens of including a three-pronged labelling (correct, incorrect and avoidant).

On **confidence and uncertainty estimation for answer avoidance**, Kuhn et al.⁴⁴ measure semantic entropy in the context of question answering, while Kadavath et al.⁴⁵ suggest that by incorporating mechanisms that allow models to assess their own uncertainty, we might develop LLMs that better understand their limitations. At present, however, models are still binary and many do not provide calibrated logprobs any more, so the situation is that we have this elimination of avoidance that we cannot easily recover. It is not one of the goals of this paper to introduce new uncertainty estimation methods or better reject rules but to highlight this loss of epistemic avoidance, understand and quantify it in the perspective of human-centred reliability, despite the existence of approaches to improve model confidence calibration or rejection. The angle from difficulty, however, can serve to improve the application of reject rules that are more in accordance with user expectations. Human study S2 is unique in making it explicit that what matters is not only how many times a model is incorrect (instead of avoidant or correct), but how often a human can mistakenly take an incorrect answer as correct. These verification overlooks depend on difficulty, something that our paper now shows very

⁴² Raimondi R, Tzoumas N, Salisbury T, Di Simplicio S, Romano MR. (2023) North East Trainee Research in Ophthalmology Network (NETRiON). Comparative analysis of large language models in the Royal College of Ophthalmologists fellowship exams. *Eye (Lond)*.;37(17):3530-3533. doi: 10.1038/s41433-023-02563-3. Epub 2023 May 9. PMID: 37161074; PMCID: PMC10686375.

⁴³ Johnson D, Goodman R, Patrinely J, Stone C, Zimmerman E, Donald R, Chang S, Berkowitz S, Finn A, Jahangir E, Scoville E, Reese T, Friedman D, Bastarache J, van der Heijden Y, Wright J, Carter N, Alexander M, Choe J, Chastain C, Zic J, Horst S, Turker I, Agarwal R, Osmundson E, Idrees K, Kieman C, Padmanabhan C, Bailey C, Schlegel C, Chambless L, Gibson M, Osterman T, Wheless L. (2023) Assessing the Accuracy and Reliability of AI-Generated Medical Responses: An Evaluation of the Chat-GPT Model. National Institutes of Health (Preprint). Not peer reviewed. Not published

⁴⁴ Kuhn, L., Gal, Y., & Farquhar, S. (2023). Semantic uncertainty: Linguistic invariances for uncertainty estimation in natural language generation. arXiv preprint arXiv:2302.09664.

⁴⁵ Kadavath, S., Conerly, T., Askell, A., Henighan, T., Drain, D., Perez, E., ... & Kaplan, J. (2022). Language Models (Mostly) Know What They Know. arXiv preprint arXiv:2207.05221.

clearly for the first time. Finally, Zhang et al.⁴⁶ present an interesting alternative to deal with answer avoidance. The authors propose to teach the LLMs to be more honest and, in this way, improve the ability of the model to answer known questions and refrain from answering unknown questions.

On **robustness to prompt formulations**, Sun et al.⁴⁷ provide insights into the robustness of instruction-tuned models and the effectiveness of prompt engineering. For its part, Sanh et al.⁴⁸ and Wang et al.⁴⁹ demonstrated the potential of multitask training and natural instructions/prompting for improving model generalisation. Prompt sensitivity is well known, but there seems to be the impression that this problem is getting better. In our paper, we analyse how robustness to different natural prompts of the same question is improved by scaling-up and shaping-up interventions and how usual prompts from regular users have an effect on performance (including avoidance), in aggregation (new Fig. A.6) and depending on the difficulty of the task (Fig. 4). All these elements are not only new but instrumental for our analysis of the evolution of LLM reliability.

With respect to other works referred to as similar **contributions** by the reviewer, our paper significantly differs from the work of Chen et al.⁵⁰, which compares GPT-3.5 and GPT-4, and whether their behaviour is changing over time, with a primary emphasis on their overall performance metrics. We do examine a wider range of models, and in particular the interplay between accuracy, item difficulty and model evolution. Our cross-task analysis provides new insights into the capabilities and limitations of LLMs, shedding light on how correctness, incorrectness and avoidance evolve and respond to varying levels of human-like difficulty. Our study also goes beyond Kojima et al.⁵¹, which demonstrates the weak LLMs' zero-shot reasoning abilities on arithmetic and other logical reasoning tasks, by a more nuanced examination of the reasoning abilities of LLMs across a wider range of reasoning tasks and how these abilities scale with increasing task difficulty. Our research also contrasts with the work of McCoy et al.⁵²: while McCoy et al. examine how the probability of the task to be performed, the probability of the target output, and the probability of the provided input affect LLM (GPT3.5 and GPT4) accuracy (on 11 tasks), our study looks at difficulty discordance, task avoidance and prompt sensitivity across LLM families. Research on the understanding and evaluation of LLMs is an emerging area that is motivated by the lack of understanding of how LLMs behave, but our paper presents a unique perspective that has no parallel in the newly introduced methodology, the non-binary metrics, the emphasis on human expectations and the perspective of family evolution through scaling and shaping up.

Suggested by R2, a relevant paper by Bansal et al.^{53,54} argues, in the context of teamwork, that AI systems should be trained in a human-centred manner, where building good mental models of the error boundaries of AI systems is paramount, and also in terms of the quality of the final decision, cost of verifying, and individual accuracies of people and AI systems. This and more needs to be brought to the analysis of LLMs.

While we cite many of these references in the paper, we discuss them in detail in Appendix A.14, because of space limitations in the main part of the paper.

⁴⁶ Zhang, H., Diao, S., Lin, Y., Fung, Y. R., Lian, Q., Wang, X., ... & Zhang, T. (2023). R-tuning: Teaching large language models to refuse unknown questions. arXiv preprint arXiv:2311.09677.

⁴⁷ Sun, J., Shaib, C., & Wallace, B. C. (2023). Evaluating the Zero-shot Robustness of Instruction-tuned Language Models. arXiv preprint arXiv:2306.11270

⁴⁸ Sanh, V., Webson, A., Raffel, C., Bach, S. H., Sutawika, L., Alyafeai, Z., ... & Rush, A. M. (2021). Multitask Prompted Training Enables Zero-shot Task Generalization. arXiv preprint arXiv:2110.08207.

⁴⁹ Wang, Y., Mishra, S., Alipoormolabashi, P., Kordi, Y., Mirzaei, A., Arunkumar, A., ... & Khashabi, D. (2022). Super-natural instructions INSTRUCTIONS: Generalization via Declarative Instructions on 1600+ NLP Tasks. arXiv preprint arXiv:2204.07705.

⁵⁰ Chen, L., Zaharia, M., & Zou, J. (2023). How is ChatGPT's Behavior Changing Over Time?. arXiv preprint arXiv:2307.09009.

⁵¹ Kojima, T., Gu, S. S., Reid, M., Matsuo, Y., & Iwasawa, Y. (2022). Large Language Models Are Zero-shot Reasoners. *Advances in Neural Information Processing Systems*, 35, 22199-22213.

⁵² McCoy, R. T., Yao, S., Friedman, D., Hardy, M., & Griffiths, T. L. (2023). Embers of Autoregression: Understanding Large Language Models through the Problem They Are Trained to Solve. arXiv preprint arXiv:2309.13638.

⁵³ Bansal, Gagan, et al. "Is the most accurate ai the best teammate? optimizing ai for teamwork." *Proceedings of the AAAI Conference on Artificial Intelligence*. Vol. 35. No. 13. 2021.

⁵⁴ Bansal, G., Nushi, B., Kamar, E., Lasecki, W. S., Weld, D. S., and Horvitz, E. (2019). Beyond accuracy: The role of mental models in human-AI team performance. In *Proceedings of the AAAI conference on human computation and crowdsourcing*, volume 7, pages 2–11.

Data & methodology

R1.3. In general, the methodology is solid although as noted above I am not convinced about the generality of the results.

Answer. See R1.1 above.

R.1.4. One small issue: the submission defines "prompt robustness" as the property that the correctness of the answer is not affected by the prompt. This implies that giving two different incorrect answers is more robust than giving a correct answer and an incorrect answer. In any case, prior work has focused on the related concept of robust accuracy (e.g., <https://arxiv.org/abs/1805.12152>), which could be generalized to recall and precision.

Answer. By prompt "robustness" we meant insensitivity to prompt changes, with the understanding that LLMs produce stable results for varying prompts. This is interpreted as the model being consistent in its behaviour, whether correct or incorrect, when similar prompts are used. We consider these two directions, especially because from the perspective of the user's expectations and the reliability of the model, it is worse when a prompt variation makes the output change from correct to incorrect than when it has always been incorrect. To have a more detailed analysis, we used prompt "robustness" for correctness and prompt "robustness" for error-freeness. Tsipras et al.'s paper relates robustness with generalisation, which could be connected to how general the LLM is for different formulations of the question (prompt). This is indirectly connected with our notion of generality through difficulty, where we expect the model to be correct for easy instances mostly, so that the changes from correct to incorrect are expected when the instances become more difficult, with the analysis of prompt sensitivity also being explored for changing difficulties. There is also an area of very difficult questions, where the model is incorrect and is expected to be incorrect, where prompt sensitivity is inexistent if we aggregate all directions of label change. Having said all this, for clarity, we have now replaced the term prompt "robustness" by "stability" throughout the paper, which doesn't seem to imply any preference towards keeping accuracy, as robustness has, and can be applied in an equally meaningful way to correctness, incorrectness and avoidance.

Regarding recall and precision, we consider three labels, and robustness is explored with correctness opposed to avoidance+incorrectness, and correctness+avoidance ('error-freeness', now 'prudence') vs incorrectness. In a 3-valued outcome, none of these metrics corresponds to precision and recall directly. The interpretation of considering precision as $\frac{\#correct}{\#correct + \#incorrect}$ and recall as $\frac{\#correct}{\#correct + \#incorrect + \#avoidance}$ is not accurate, and would be very confusing, since precision and recall need a full matrix of four values and we only have three, as we do not have a split of avoidances into those that would go to correct or incorrect. That is why there are several extensions of precision and recall metrics when using reject rules⁵⁵, as they should not be used in reject rule or three-valued scenarios. Nevertheless, we mention the issue of dealing with 3x3 confusion matrices in section 4.4 now, and in appendix A.14 we now discuss the traditional terminology (and the tension between precision and recall, and the tension between accuracy and robustness, as suggested by the reviewer) within our methodology explicitly. (see more about terminology in R1.7 below)

Appropriate use of statistics

I have no concerns

Conclusions

⁵⁵ Fischer, L., & Wollstadt, P. (2023). Precision and Recall Reject Curves for Classification. arXiv preprint arXiv:2308.08381.

R.1.5. As noted above, I accept the conclusions within the scope of the tasks presented here, but I'm skeptical about the generalization to more typical usage cases.

Answer. As discussed in R1.1, we have fortified our study with dozens of new tasks that closely mirror real-life user interactions with language models according to several studies (see R.1.1). These new tasks are now organised into two benchmarks (**transforms** and **science**) and the methodology, especially the difficulty metrics, is applied to multi-task benchmarks rather than single-task datasets as before. Also, we now have different levels of open-endedness and a diversity of domains. All this totals more than four million LLM interactions, considering the (approximately) 9200 instances, 15 realistic prompts each and 32 models. The results are consistent in the range of open-endedness and for various domains, which supports the generalisability of the findings.

Improvements

R.1.6. In general, the paper does not engage very much with the prior literature on LLMs, mostly citing this literature through surveys (e.g., Zhao et al 2023; Yang et al 2023; Liu et al 2023). The paper would benefit from engaging more with prior work on LLM calibration and confidence, robustness, and change over time.

References

On LLMs and their weaknesses at mathematical reasoning: <https://arxiv.org/abs/2205.11916> is just one example, which evaluates text-davinci-002 on six mathematical reasoning datasets.

On string manipulation:

<https://arxiv.org/abs/2309.13638>, which is also a useful reference in regard to item difficulty, because it shows how GPT 3.5 and GPT 4 error rates relate to properties of the autoregressive pretraining task.

Some additional citations related to robustness to prompt reformulations:

<https://arxiv.org/abs/2110.08207>

<https://arxiv.org/abs/2204.07705>

On LLM accuracy versus difficulty for humans:

Comparative analysis of large language models in the Royal College of Ophthalmologists fellowship exams (<https://www.nature.com/articles/s41433-023-02563-3>)

Assessing the Accuracy and Reliability of AI-Generated Medical Responses: An Evaluation of the Chat-GPT Model (<https://www.ncbi.nlm.nih.gov/pmc/articles/PMC10002821/>)

Answer. See R1.2 above where these references are already covered. Given the limitations on the number of references in the main part of Nature papers, we referred to surveys often. We have now expanded our literature review to include both seminal papers and recent studies that have significantly contributed to the understanding of LLM calibration, confidence estimation, robustness to input variation and model evolution. Still, because of space limitations, we do most of this review in appendix A.14, where we cover all the references above and many more that have appeared before and after our submission, and in the past few months as well. In particular, we cover the references suggested by the reviewer, grouped into several blocks. First, we cover techniques for model confidence estimation. Second, the robustness of LLMs to different types of prompt variations. Finally, we cover the partial longitudinal studies that track changes in model capabilities, biases, and errors over successive releases.

Also, we have included the reference to Kojima et al.⁵⁶ (mathematical reasoning of text-davinci-002) to provide a historical perspective on the development of these models. The additional citations on robustness to prompt reformulation have been essential in situating our observations within the broader discourse of LLM development. Finally, we have drawn on the suggested medical-based studies to highlight the relevance of our work in domains where the reliability of AI-generated responses is critical.

R.1.7. *Clarity and context*

Figure 1 and Table A.7 (now A.9) are quite difficult to understand. Part of the issue is terminology: the top and bottom parts of the figures are labeled "useful/safe" and also "correctness/error-freeness" but actually I think these concepts are better described by well-known existing terminology of recall and precision.

The mathematical notation for correctness is also non-standard and in my opinion unintuitive -- \oplus and \ominus are more typically treated as generalized addition and subtraction operators. The relationship of correctness and error-freeness to "prompt robustness" and "difficulty concordance" was not understandable from the caption; I have concerns about the definition of robustness that I mentioned above.

I didn't understand if the difficulty in the "locality" question is related to the city in the question or in the answer. Both seem relevant.

Answer. As explained above, the use of the terms precision and recall would be misleading for correctness, and especially for what we called error-freeness, since it considers correct + avoidance, and avoidance cannot be considered as part of recall. This error-freeness is what matters for reliability, more than correctness and more than the %times the system doesn't avoid the question. For the same reason we cannot use the terms specificity or sensitivity. Actually, we have three possible labels with a 3x3 confusion matrix instead of a 2x2 binary confusion matrix, so we should not use any of the standard terms in binary classification or retrieval.

We had long internal discussions about the terminology and checked the literature for new terms (e.g., the term "honest" in the new triad of honest, harmless and helpful terminology, see e.g., Askill et al.⁵⁷), but the fit is imperfect and would be misleading. We are open for suggestions but in this new version we simplify the terminology by keeping "**CORRECTNESS**" (which is simply capturing the correct label against avoidant and incorrect) and replacing error-freeness by '**PRUDENT**', which denotes aversion to error (not being wrong, capturing the correct and avoidance levels as opposed to incorrect). For clarity we also avoid the terms useful and safe now for these two categories. With this change and a rewriting of the caption of those figures and accompanying text, it is now more clear.

We have replaced the \oplus and \ominus symbols with a more traditional functional notation, using c , a and i as Boolean functions for correctness, avoidance and incorrectness, respectively. This is also used in the figures, captions and definitions in other parts of the paper for better understanding, including Fig. 1 and Table A.9. Also, as said in R.1.4, we have renamed prompt robustness as stability. The relationship between the different metrics is now explained in a more straightforward way, and the terminology is revisited in appendix A.14. Similarly, clarity around "prompt stability" and "difficulty concordance" has been improved with additional context in captions and elsewhere. Finally, the difficulty of the locality benchmark has been clarified: it indeed considers both the input and target city (the correct answer).

⁵⁶ Kojima, T., Gu, S. S., Reid, M., Matsuo, Y., & Iwasawa, Y. (2022). Large Language Models Are Zero-shot Reasoners. *Advances in Neural Information Processing Systems*, 35, 22199-22213.

⁵⁷ A. Askill, Y. Bai, A. Chen, D. Drain, D. Ganguli, T. Henighan, A. Jones, N. Joseph, B. Mann, N. DasSarma, N. Elhage, Z. Hatfield-Dodds, D. Hernandez, J. Kernion, K. Ndousse, C. Olsson, D. Amodei, T. Brown, J. Clark, S. McCandlish, C. Olah, and J. Kaplan. A General Language Assistant as a Laboratory for Alignment. arXiv:2112.00861 [cs], Dec. 2021. arXiv: 2112.00861.

Reviewer 2

Large language models (LLMs) like ChatGPT have captivated public attention. The ease by which these systems generate natural language, converse with users, and seemingly solve a wide-range of problems has teased the potential that they can radically transform many domains – from the ways students learn to how patients interact with doctors. Yet, how reliable are these systems for such users? In this work, the authors explore the reliability of a suite of LLMs – focusing particularly on the impact of two trends in LLM development: scaling these models, and "shaping" them (via instructions and human feedback). The authors consider three varied model families (GPT, LLaMA, BLOOM), over three tasks (arithmetic, anagram puzzles, and trivia-style locality questions), along three different dimensions of reliability (predictability according to difficulty level, tendency towards avoidance behavior, and robustness to prompt alterations). From their empirical investigations, the authors conclude that shaped up models have unpredictable error patterns, which raises caution for users and developers of these systems alike.

First, I applaud the authors for calling attention to this important problem: updating and scaling these models does not guarantee their reliability. Ensuring predictable reliability is paramount for human users to build good mental models of when they can be appropriately used [1, 2]. I believe the authors' study, which frames and brings this issue to light, will be of interest (or certainly ought to be of interest) to researchers and practitioners across many disciplines – as this new age of LLMs is poised to touch all. I also find the authors' delineation and disentanglement of two trends of LLM development ("scaling up" and "shaping up") clarifying and think it can help contribute positively to the collective language around how we describe these models and their evolution when discussing how such systems ought to be developed, deployed, and regulated. Further, I believe the metrics the authors explore reliability according to — "difficulty concordance", frequency of avoidance, prompt robustness – are in principle valuable dimensions to evaluate. Taken together, the unpredictable landscape of errors that these authors find, and the intriguing large decrease in "avoidant" responses on these rote tasks, are interesting observations and may surprise some readers. I also applaud the authors on considering a varied suite of model families; the choice of GPT, LLaMA, and BLOOM are all sensible and valuable to offer deeper investigation into (especially GPT and LLaMA given their popularity, as the authors note).

However, I maintain several concerns about the paper in its current form. My concerns fall into three interrelated buckets: concern about the difficulty proxies, the task selection more broadly, and the interpretation of the results. I address each in turn.

Difficulty measures:

R2.1. First and most pressingly, I have deep concerns about the validity of the authors' difficulty metrics. Based on my understanding of the authors' description of their methodology in Section 4.1 and Appendix A.4, the authors curate and create a list of possible proxies of human difficulty for their tasks and then only use the difficulty metric which is most correlated with final performance. To quote the authors: "in the end, as all of them are created to be human-like, we chose the most predictive" (line 309-310). I find this method highly problematic. Selecting a metric so that it is predictive, and then claiming that a metric is a valuable predictor (claim F1a in Section 3, line 11, and line 123, for instance) is circular.

Answer. We selected or devised all metrics to be human-like in the first place, as what we want to explore is human expectations. But we agree this is problematic if we don't double check that they are actually good in terms of human expectations. Accordingly, we have now conducted a human study S1 with a sample of 90 instances per benchmark and 5 respondents each (see responses **E.2** and **R2.6** for more details), where we now choose the difficulty metric that is most negatively correlated with human expectations about success on

the task (all correlations with p-values well below 0.05, see the chosen metrics, in bold, in Table A.8). We then use these chosen metrics as proxies for all items in the benchmark. This new procedure has led to some changes in the x-axis representing difficulty as well as the aggregated metrics. In particular, `f_scb` is now replaced with `f_let` in **anagram** as the chosen metric. This makes sense as the number of letters in an anagram is the first thing humans pay attention to when estimating difficulty. For **addition** and **locality**, the metrics that correlate most with human expectation are `f_hrm` and `f_pop`, the ones we used in the previous version, so there is no change in this domain. For the newly introduced **transforms** benchmark, we proceeded in the same way: we figured out possible analytical difficulty metrics that seemed human-like, and then we evaluated which one correlates the most with human expectation. For this new benchmark, it was `f_{w+l}`, a combination of edit distance and length, which is used for the plots and the indicators. Finally, for the **science** benchmark we already had several sources of human difficulty for all the instances, so we used the human study S1 to calibrate them in the same scale given by the survey population. The set of old and new metrics is explained in detail in appendix A.5, the correlations and the justification of the choice in Table A.8, and details of this study and their questions in new Appendices A.10 and A.11.

R2.2. A quick look at Table A.6 (no longer in the new version) reveals that many of the difficulty metrics the authors originally considered are actually not particularly correlated with model performance (especially for the BLOOM family). In fact, some difficulty measures have zero (!) correlation with performance. My understanding is that these correlations (in Table A.6) (no longer in the new version) are over the same test instances used in the rest of the paper?

Answer. The observation about the correlation with BLOOM's performance in old Table A.6 is correct; the largely low correlation across difficulty metrics is expected due to the low performance of many models in this family, and the results are very poor for many tasks and difficulties. As soon as a model has some more correctness, the correlations appear.

Regarding the other models, except for `f_inp`, which is simply a bad metric, all difficulty metrics actually showed significant correlation with model performance. The values were calculated as a mean of correlations for all models in the family and we now calculate the correlation for all data in the family for the five difficulty functions that were chosen. Now, the lowest correlation is -0.43 for BLOOM and arithmetic, but all the other correlations are high (Table A.10). But now, thanks to the reviewer's suggestion, we do not use the metric that shows the highest correlation with the language models, but the one that is most correlated with human expectation (Table A.8). With this procedure, any range of *model* correlation with difficulty is meaningful: when the correlation is low this may be due to the model being very poor or being in strong discordance with human expectations, while high correlations represent the model is very competent on those instances human consider easy while being clearly less performant where humans expect more failure.

The understanding is correct that the correlations presented in (old) Table A.6 (now replaced by a similar Table A.10) derive from the same test instances as those used throughout the paper. For the new correlations between the metrics and human expectation (S1), found in Table A.8, we used a sample of instances, the ones that are included in the human study S1. But for the correlation between model performance and the metrics (Table A.10), we use all data. In general, as guessed by the reviewer, the results are for all data, unless otherwise stated.

Apart from the new tasks and the new procedure for choosing the difficulty metrics, we have improved the clarity in our presentation about the factors that affect difficulty in Section 4.1 and new Table A.7.

R2.4. As such, I believe selecting the metric which is most predictive and then claiming that the correctness can be predicted by difficulty is effectively cherrypicking. Unfortunately, many of the analyses in the paper are based on sorting the tasks by difficulty, using the effectively-cherrypicked metric per task, which worries me about how much can be drawn about correlations and trends in the other metrics that fall out of the difficulty sorting.

Answer. The process we followed aimed to identify difficulty metrics that represented the demands of the task at hand, rather than arbitrarily "cherry-picking" metrics to suggest predictability where there may be none. Our aim was to show that with an appropriate difficulty function—one that effectively encapsulates the factors that make each task instance inherently complex for a language model—we can indeed observe predictable patterns of correctness, avoidance and incorrectness. All difficulty metrics in the paper (except those that are used in **science**) are defined by rules that exclusively use the instance text, and can be applied for any future new instance. Looking for features that actually predict performance is something that humans do to build their models of what is difficult and what is easy, and our paper must build on that. Having said this, we acknowledge that once we define them by identifying the factors that affect the difficulty of the instances (following the literature or common sense), we shouldn't have chosen the one that is most predictable for the language models we want to explore, because of the risk of overfitting to these models.

As said above, we have now made the final selection using human perceptions of difficulty from study S1. By including a human study to substantiate our choice of difficulty metrics, we confirm that they correspond to how challenging questions look for humans across the breadth of tasks.

R2.5. Instead, it may have been more appropriate for the authors to aggregate the correlations across all considered difficulty measures.

Answer. Thanks to the new human study S1 (appendix A.11), we now have evidence that we should not take all difficulty metrics as equally valid and representative of human expectations. On the one hand, the use of a *very predictive* difficulty metric that is not negatively correlated with human expectations tells us little about the reliability of language models, since humans do not have that metric in mind when asking the models or accepting their answers, something that we explore in a second human study S2 (see appendix A.12). On the other hand, the use of a *non-predictive* difficulty metric that is not negatively correlated with human expectations is basically useless for a reliability study. We now analyse the factors that should affect performance in more detail (see new Table A.7) and include more difficulty metrics combining these metrics in more ways. But some of these combinations are simply not good as proxies for difficulty and it would not make sense to average them. This is why we now select the metric that is most correlated with human expectations (against questions Q1 and Q4 in S1). Also, through this study S1 we can also check that *if it is correlated with human expectations*, then it is also correlated with actual human performance and self-confidence (questions Q2 and Q3 in S1). To complete the analysis we not only include the correlations of all the considered metrics with human expectations (Table A.8), but also between them (see Figure A.14), which suggests that changes would be minimal if we made a different choice of the questions.

R2.6. Additionally, it is not clear to me that the difficulty measures the authors choose for the locality metric actually correlate with real human difficulty (which seems, in part, to be the authors' claims);

Answer. Finding an intuitive and simple difficulty metric for the locality task is challenging, as this is a task that has an invariant formulation that depends on geographical and cultural factors. We intentionally wanted to consider a difficulty function that could integrate more syntactic and semantic elements.

The correlations from study S1 for locality confirm that some of the difficulty metrics for locality have low correlation with human expectations. For instance, only considering the frequency of the cities is not enough (f_{inp} and f_{tar} with correlations with Q1 of -0.06 and -0.20 respectively and with Q4 of 0.08 (positive) and -0.20 respectively, as shown in Table A.12). Joining Q1 \cup Q4 leads to 0.02 and -0.22 respectively. We explored

more metrics in this domain. We consider the population of the cities or countries (f_{cip} and f_{cop}), or the distance between cities, as suggested below, on its own (f_{dst}) or combined with f_{pop} (resulting in f_{all}) is also insufficient (with correlations with Q1 U Q4 of -0.39, -0.34, -0.17 and -0.46 respectively). However, the metric we used originally in the previous version of the paper, f_{pop} , shows correlations of -0.51 with Q1 and -0.46 for Q4, and stays at -0.47 for Q1 U Q4. It's not as high as in the other tasks, but still reasonably good for a population that might be culturally biased, and a task that may depend on many factors. Also, this is strongly affected by the high perceived difficulty of this task by humans according to S1.

R2.7. I'm surprised none of the submetrics the authors consider incorporate the distance value, alongside information about the cities, as I would think that could impact human difficulty? I hesitate then to call this particular metric a "human difficulty metric" (line 96) without cross-verification that it does correlate with humans' perceived difficulty. However, I am not an expert on the actual difficulty measures. My principal concern here is the selection of the metric which is most predictive, and then claiming throughout that correctness tracks (or doesn't track) with difficulty.

Answer. This comment from the reviewer was critical to embark ourselves in the human study presented above, which actually performed this verification. This also allows us to check very easily if something like city distance has any correlation with the difficulty expectation from the human study. The correlation of the difficulty function using only distance (f_{dst}) with Q1 and Q4 is low, -0.17, but combined with some other features of the instance (f_{all}) can lead to stronger correlations, up to -0.46 in Q1 U Q4, not showing a significant advantage over the original difficulty metric (Table A.12).

Tasks

R2.8. In addition to my concerns about the difficulty measures used/selected which are unfortunately quite endemic to the paper given how most analyses build on the difficulty-sorting (see Fig 2, x axis), I also have some concern over the tasks considered. The authors pitch the paper as investigating the "user-driven reliability" (line 48) and usefulness of families of LLMs. The authors justify their tasks as being more realistic to how real users will engage than the multiple-choice questions that often make up LLM benchmarks. Yet, the tasks that the authors consider are quite contrived, in my opinion; with the exception of arithmetic, which is most clearly important to downstream user needs (say, if used in any mathematical setting), it is less clear the importance of the anagram and locality tasks. As such, I again have caution in the generalizability of these findings. How do these reliance measures track across other, real-world domains (especially of the kinds the authors seem motivated by?) And critically, how do they track against real human user beliefs and experiences? The authors note in lines 207-209: "we are urged to analyse how reliable these models are according to the expectations of a regular human user, prompting the system in natural ways," yet no human users are consulted or incorporated in this study?

Answer. Our intention was not to exactly mimic real-world domains, but to represent broader cognitive domains such as numeracy, vocabulary and spatial reasoning with carefully chosen tasks such as addition, anagrams and location respectively. However, we understand the caution to overgeneralise from just three core tasks. In this revision we have extended the analysis to cover a wide range of practical scenarios through dozens of different tasks that users may encounter more frequently (see **R1.1**). These tasks include several common text and data transformations, scientific questions in several domains and complex queries with open-ended inputs and outputs involving hundreds of characters. These new tasks more closely reflect real user interactions with LLMs.

Importantly, our revised methodology now incorporates feedback from human users (see the discussion of study S1 in **R2.6** and described in appendix A.11 in the paper) to determine the difficulty function, as described above. But this study also collects actual difficulty and self-confidence of human users when facing these questions. Study S2 (see appendix A.12) gets key information about reliability by asking humans to assess the outputs that come from LLMs, mimicking a situation of humans being verifiers or supervisors in which they

have to determine if they accept or reject the output coming from the (machine) “assistant”. We analyse this for several types of errors (especially when the output of the model is incorrect and is labelled as correct by the human) and explore the effect of difficulty on reliability (see **E.2** and new figures A.15 to A.19 in the appendix).

Regarding our criticism of multiple-choice benchmarks but the inclusion of **science** in this version, the reason was actually motivated by investigating whether close-ended benchmarks show a different behaviour. As we mention in response E1, the results are similar in general, but the mostly correct low-difficulty region for **science** with GPT-4 (see Figure 2) suggests that multiple-choice benchmarks may be giving a distorted (too-optimistic) view about the reliability of language models.

R2.9. This is not to say that understanding model behavior on these kinds of tasks, without external human validation, is not valuable; however, for the authors to draw broader conclusions about whether we should change the ways we scale up / shape up these models, I think that more realistic and open-ended tasks of the kinds that humans may actually use these systems for are needed. Ideally, the authors would include at least one more open-ended domain that is more realistic of how users may actually interact with these models (say, asking for help on some educational tasks that relies on “general world knowledge”).

Answer. The first new benchmark, **transforms**, contains hundreds of instances comprising 73 different tasks in many different domains, reflecting complex, real-world operations that users commonly perform with language models, such as data extraction, summarisation and conversion between formats. The second new benchmark, **science**, is composed of educational Q&A items testing general scientific knowledge, and more advanced questions in specific domains.

It is important to emphasise that **transforms** is a collection of tasks for which we use the same difficulty metric for all of them, which is based on some characteristics such as size and syntactic distance. From S1, as we now do with all other metrics, we show high correlations, which is significant taking into account the diversity of tasks in this benchmark. Finally, the **science** benchmark has human difficulty functions for all items, and we use S1 to calibrate difficulty coming from different populations.

R2.10. As the authors note, two issues may arise: the difficulty measure may be less clear (but, as mentioned above, I think the current method for selecting such measures is flawed), and the open-ended nature of the responses may be harder to evaluate. I do not think the latter should prohibit the authors from exploring a more ecologically valid task; for instance, a crowdsourcing study could be run to recruit humans to evaluate. While of course this would take time and could potentially get expensive, I do think some aspect of external human user studies is important to sharpen the claims made in the paper about how scaling-up / shaping-up impact “user-driven” reliability.

Answer. Study S2 was devised following this comment. While evaluating creative tasks would be too subjective, and would test user satisfaction rather than the actual accuracy of the results, we instead evaluate tasks having ground truth. The advantage of **transforms** and **science** is that they have objective answers, so we can not only ask humans to give answers to these questions (in S1) and grade them objectively, but also ask humans to assess the answers from language models, as we do in S2. Note that for all five benchmarks in the current version of the paper, including the multiple-choice questions, both humans and LLMs can respond in an open-ended way, with free text, which is key for the way avoidance can manifest itself.

S2 shows that crowdsourced humans are generally bad at assessing these questions, and make many critical mistakes such as taking incorrect answers as correct (see new figures A.15 to A.19). This reinforces the motivation of studying reliability, as well as the effect of difficulty and a three-pronged analysis of answers (correct, incorrect and avoidance).

R2.11. Inclusion of at least one open-ended domain would also permit a more nuanced investigation to the impact of avoidance. I do not believe the current paper has sufficient discussion around the proportion and type of avoidance behavior that may arise in other safety-critical user-facing domains like education, medicine, etc, where we may want it... I would hypothesize shaping-up may increase avoidance behavior frequency, not decrease it... (though this would be good to explore!)

Answer. The addition of a wide range of questions from multiple-choice to open-ended transformations in this new version allows us to understand avoidance better and strengthen the support of our previous findings. We see similar behaviours for avoidance in tasks of different character and domains, corroborating the major finding that shaping-up is actually reducing avoidance, for all kinds of tasks and domains (including tasks in the medical and educational domains coming from the two new benchmarks). This is very clear for the non-conforming and epistemic avoidances, while marginally surviving for ethical avoidance (see Figure 3). We also thought there would be more avoidance of all kinds, even more than the level we find in the ethical category, due to all the shaping-up mechanisms such as RLHF, but what we find is very consistent across all formats, domains and models. Either developers are not realising the problem of lower avoidance or they want to maximise %correct at all costs. This may be strongly influenced by the tradition of evaluating LLMs and many other machine learning models as correct/incorrect only. The discussion section has been modified to emphasise this vanishing of avoidance in only a couple of years, also mentioning that specialised language models in medicine and other critical areas may be designed with reject options, so favouring avoidance.

R2.12. Secondly, focusing just on the tasks that the authors selected — I actually do think that the tasks are confounded (line 252)? In particular, both the Addition and Locality tasks rely on an ability to “understand” and work with numbers. As such, it is possible that failures across tasks can be attributed to issues of these models with numerosity. It would be good for the authors to briefly discuss this.

Answer. While we tried to select the original three domains to represent core tasks in three different clusters, it is very difficult to build “pure” tasks, and all of them require some language capability (to understand different prompts) and some of them share some other more specific capabilities, such as Addition and Locality (the first more loaded on arithmetic, but the second requiring distances, and both using numbers in the formulation). With the inclusion of the new tasks we now have a much less pure group of tasks and benchmarks. We now include a new Table A.7 in Appendix A.5, where we distinguish the primary and secondary abilities involved in each benchmark, with all benchmarks relying on language understanding as indicated by the reviewer, and locality relying on numerical ability as well. We use this table as well to identify the factors that, depending on the task, may affect difficulty.

Interpretation of the results:

R2.13. Finally, and perhaps easiest to address, I feel that the authors’ interpretation of the results neglects to adequately disentangle performance along the dimensions the authors nicely laid out in their Introduction: not just that these models are being shaped up, but that they are being scaled up. The authors focus primarily on the impact of shaping up these models; understandably, this seems to have the biggest impact on the performance metrics considered. However, what about the impact of scaling? For instance, if you look directly at the LLaMA instances and compare models with or without shaping up, but different parameter sizes (e.g., LLaMA-2-7b-chat vs LLaMA-2-13b-chat vs. LLaMA-2-70b-chat). Such investigations can help researchers gain deeper insight into the impact of scaling, and I felt not adequately digging into such results was a lost opportunity in this first paper version. Again, though, I feel that before deeper analyses can be done along those lines, the issue of the difficulty measures ought to be addressed.

Answer. In this revised version, we have included a new section (2.4) where we analyse the effect of scaling, specifically using the paired raw-chat and raw-instruct variants of the LLAMA and BLOOM families respectively, i.e., LLAMA-2 (7b, 13b and 70b) and BLOOM (560m, 1.1b, 1.7b, 3b, 7.1b, 176b). As the training data, architectures and many hyperparameters for these two sets are equal, we can limit the effect of

confounding factors in our analysis of scaling (also compute, as it is linearly related to size, see response R2.21 below). This is a great advantage of LLAMA and BLOOM for this analysis, as this is not possible for more popular and fully commercial families such as GPT, not having a dual evolution with and without shaping. Also, for GPT, we do not have full information about the scale, and several changes are performed with new versions of the models at the same time, acting as confounders.

In this ablation with LLAMA-2 and BLOOM, we take a scaling-laws approach. Our results confirm that, as expected, scaling does indeed contribute positively to certain aspects of model performance, such as a general increase in the proportion of correct predictions as model size increases. However, incorrectness does not decrease, and for the raw models, it increases significantly. This is more evident when we analyse the percentage of incorrect responses among all responses that are not correct, $i/(a+i)$ in our notation. Here we see an alarming increase of the proportion of errors, with models becoming more “ultracrepidarian” (increasingly giving a non-avoidant answer when they do not know, so failing proportionally more).

Although scaling laws have been applied to many models and datasets in the past, this is the first time, to our knowledge, that a scaling laws analysis is performed with a three-pronged labelling of items, separating correctness, avoidance and incorrectness. We thank the reviewer for this suggestion, which has been incorporated as a new subsection in the paper (2.4).

R2.14. Overall, given these limitations, I think that it is not appropriate to draw – from these current results – that we need a change in the direction of scaling up and shaping up these models (lines 249-250). I think adequately addressing such claims needs at least one more open-ended task, and ideally, the inclusion of real humans “in-the-loop” of evaluating the output. Again, I believe that the high-level problem statement and research questions that the authors lay out are of value and interest to many research communities. I also applaud the authors on their granular de-aggregated analyses. I certainly found the piece interesting and believe that, if executed rigorously with appropriate distance metrics and ideally an additional ecologically valid task, the work could bring value to many communities. I also reiterate that it is no easy feat to evaluate this many models, over this many problems, and admire the authors’ courage to take on that task.

Answer. As we have mentioned in the responses above, we have included more open-ended tasks that better represent realistic and practical user interactions with large language models. In addition, we have incorporated real human evaluations into our methodology by including the crowdsourced study S2. This allows us to capture authentic user perspectives on the use of these model families, providing a more holistic assessment of their reliability and effectiveness in real-world scenarios, where the questions users ask and the prompts they use are affected by expectations on difficulty, as so are the verification and supervision procedures, with the extra element of avoidant answers. We now have increased support to suggest a change in the direction of scaling up and shaping up as we do in the discussion section. The results from S2 also add support to our recommendations of increasing difficulty concordance, and, at least, epistemic avoidance.

Specific/Minor:

Finally, I leave a few specific and minor comments/questions that I hope will improve a future iteration of the authors’ work:

R2.15. I am confused as to what this “human expectations” dataset would encompass (Section 3). Expectations of what? Model behavior?

Answer. We have rephrased this in section 3, as we now have two studies partly covering “human expectations”, one about the difficulty of a task before or independently from using a language model, hence representing the expectations that humans themselves or other humans may have on solving the tasks correctly (the difficulty estimations), and a second one about the expectations of a model that is correct/avoidant/incorrect is assigned to a different category when humans inspect these outputs

(supervision/verification). Other datasets about the expectations humans have about a particular language model require a longitudinal study where users become familiar with the tool and learn a mental model of what the LLM can do, before asking them.

We plan to leave all the results from the human studies available, but these are not meant directly for finetuning and RLHF. Their size, and the phrasing of the questions, is meant for double checking the findings and takeaways of our study. This is why we call for collective datasets that go beyond preferences and human satisfaction (such as LMSYS Chatbot arena and AlpacaEval), but include human expectation questions as the ones we have collected in our studies, and more.

R2.16. I appreciate the authors' avoidance taxonomy and found it helpful. However, I want to raise attention that the example given for C1 does include a first-person singular pronoun? I found the definition therein confusing and perhaps contradictory. Additionally, I found the delineation of the impact of certain forms of avoidance arising from "shaping up" as also perhaps a bit imprecise; shaping up may change avoidance behavior in ways beyond getting the model to say "I am not a language model." Shaping up could also, in other domains – say that are more open-ended and ecologically valid as I touched on above – could lead to more of the kind of avoidance without first person "I". I'd encourage a renaming of that part of the categories.

Answer. The example provided for C1 was indeed inconsistent with the definition, and has been corrected. Shaping up may contribute to more sophisticated forms of avoidance, particularly in open-ended or real-world tasks where models may display avoidance behaviour that does not use first-person pronouns, self-identifies as an individual or mentions content policy explicitly. We have renamed "spontaneous" and "by shaping" as passive and active respectively. This separates the confusing association with raw and shaped-up models respectively, as this is something we cannot know for certain (which we now say explicitly). Table 3 now includes more examples and labels them as passive or active. Actually, we found some active non-conforming ones.

R2.17. When was GPT-3.5-turbo accessed? It would be good to add the date(s). I believe that this model may be continually updated by OpenAI. The updates can matter [3]. I appreciate that the authors listed the exact versions of GPT-4 used.

Answer. We used the GPT-3.5-Turbo model with build GPT-3.5-0301 (March 2023), for the old and the new benchmarks. This information has been added in section 4.5. The models in the GPT family may suffer updates that are not always reflected in their build dates. In fact, we include two different builds of GPT-4 as a way of showing the variability that may exist due to updates. This is another reason to analyse and keep track of open-source families such as LLaMA and BLOOM.

R2.18. Additionally, and a minor note: it would be good for the authors, for reproducibility, to note in the Supplement what mode of interaction they used with the various APIs, particularly GPT. For instance, did the authors use the Completion or ChatCompletion APIs? Was a "system-level" prompt provided; if so, what was it?

Answer. For the GPT family of models, we used the commonly used ChatCompletion API⁵⁸ to query the models, as it is designed for conversational interactions and allows for a more dynamic exchange, which is consistent with our research goals. We did not specify any system-level prompts to avoid the introduction of any potential biases. All settings and parameters used with the APIs, such as temperature, max tokens, stop conditions, etc., are now included in section 4.5.

R2.19. The authors source prompt templates from pre-existing textbook / web links. As such, it's possible – I'd say highly probable – that these text snippets from which prompts are derived have been included in the

⁵⁸ <https://platform.openai.com/docs/api-reference/chat/streaming>

web-scraped training data of these models. Are there trends the authors find in the kinds of prompts that models are least robust to? Such an analysis could be nice, but is not necessary to the overall story.

Answer. We share the reviewer's concern about sourcing prompt templates from existing textbooks and web sources. Even if we didn't copy the prompts verbatim but introduced variations, there is a high likelihood that these snippets are very much like the extensive web-scraped datasets used to train these models. So the reduction of prompt sensitivity may be partly affected by this contamination phenomenon.

For the **science** benchmark we now borrowed from the sources suggested by reviewer R1 (see **R1.2**). In this case, we reuse prompts as we want them to be ecologically valid, and also help with the comparison of previous results. But, this must assume that some degree of contamination may have happened in the latest models, either during training or after it, through prompt leakage⁵⁹.

However, for the **transforms** benchmark we introduced an original variation of prompts, with five major classes:

- Pattern: the task is explained with general patterns or expressions;
- Constraints: the task is explained with the constraints the output must meet;
- Algorithmic: the task is explained as a procedure or with some steps;
- Denotational: the task is explained using names or references to other concepts; and
- Illustrative: the task is illustrated with some partial cases or examples.

Then, for each of these five classes, we have three variants, where the input (the key part that makes different instances of the task) is placed at the beginning, in the middle or at the end of the prompt. Given the high diversity of the tasks of this benchmark, the different variations for each task were defined specifically and manually for each transformation. All this has been added to Appendix A.2.

In general, we feel there's a tension between making highly common yet contaminated prompts and less natural yet fresh. A rigorous study of contamination on prompt sensitivity should be a full paper in itself.

But we have added a new analysis of prompts at the end of appendix A.2, Figure A.6, showing different models and benchmarks binned by each of the 15 prompt templates. For GPT, we see instability in the early models (Ada to Davinci) and a bit for GPT-3.5-Turbo for locality. For LLaMA and BLOOM, we see that stability is much lower, with peaks and valleys for different prompts, mostly to radical changes in avoidance (from 0 in some prompts to 100% in some others). Looking at the benchmarks, we find that the **transforms** benchmark, which uses new prompts, shows more stability than the **science** benchmark, which reuses prompts from the literature. Looking at the time some of the models appeared, we cannot really extract any clear contamination pattern. All this is now discussed at the end of appendix A.2, supported by Figure A.6.

R2.20. I am confused on the authors' conclusions about what should be done to improve reliability. I hesitate to draw the conclusion that training "should" give more relevance to easy problems; it's not immediately clear that this will improve the (lines 224-225). And lines 228-230 call for just more shaping up (fine-tuning)?

Answer. We have rephrased this part of the discussion section to emphasise that scaling up and shaping up could potentially take models in the right direction if the loss functions were modified, both during and after training time. Scaling up could have more effect on difficulty concordance, while shaping up, through fine-tuning, RLHF or other methods, has much more potential for adjusting avoidance and increasing prompting stability. But the new text doesn't say exactly how this has to be done, but simply that this has to be done differently from now on, changing the loss and reward functions such that difficulty concordance

⁵⁹ Balloccu, S., Schmidová, P., Lango, M., & Dušek, O. (2024). Leak, cheat, repeat: Data contamination and evaluation malpractices in closed-source LLMs. arXiv preprint arXiv:2402.03927.

increases (errors in easy instances are penalised much more than in hard instances) and prudence indicators improve (incorrect results are penalised much more than epistemic avoidance).

R2.21. Lastly, the authors discuss three kinds of "scaling up" in the introduction – number of parameters, amount of training data, and training time. To my reading, the authors do not discuss the impact of training time at all? Nor do the authors break down the impact of scaling along the number of parameters vs. amount of training data. This is fine, but it could be worth noting that the different forms of scaling up could be conflated and may have differential impact on reliability that this study is not able to address (but are ripe grounds for future work).

Answer. This is a good point that we overlooked, because the analysis of scaling laws usually takes for granted that these three elements are chosen so that none of them is a bottleneck, following empirical laws determining the optimal number of parameters (model size) and number of tokens (data size) given some available compute (Hoffman et al. 2022, the "Chinchilla" paper⁶⁰).

Actually, we have investigated the relation between the model size, as explored in the new section 2.4 of our paper, and compute. From <https://llama-2.ai/llama-2-model-details/>, we got this results:

MODEL	#Parameters	#Tokens	GPU hours
Llama 2 7B	7B	2.0T	184320
Llama 2 13B	13B	2.0T	368640
Llama 2 70B	70B	2.0T	1720320

We did not find the compute information for BLOOM at the level of each particular model (<https://huggingface.co/bigscience/bloom>, <https://arxiv.org/pdf/2211.05100.pdf>), other than the reported number of training time for all of them as a whole (1M GPU training hours) <https://medium.com/codenlp/the-training-time-of-the-foundation-models-from-scratch-59bbce90cc87>.

However, from the table above, we see that the #parameters and GPU hours are linearly related, so we hypothesise (because of this and the choice of parameters to fit data and compute following the Chinchilla paper) that our analysis for number of parameters in section 2.4 would be redundant with a similar analysis for compute.

R2.22. (References suggested by the reviewer)

[1] Bansal, G., Nushi, B., Kamar, E., Lasecki, W. S., Weld, D. S., & Horvitz, E. (2019). Beyond Accuracy: The Role of Mental Models in Human-AI Team Performance. Proceedings of the AAAI Conference on Human Computation and Crowdsourcing, 7(1), 2-11.

[2] Bansal, G., Nushi, B., Kamar, E., Horvitz, E., & Weld, D.S. (2021). Is the Most Accurate AI the Best Teammate? Optimizing AI for Teamwork. AAAI Conference on Artificial Intelligence.

[3] Chen, L., Zaharia, M., & Zou, J. (2023). How is ChatGPT's behavior changing over time?. arXiv preprint arXiv:2307.09009.

Answer. Thank you. The papers by Bansal et al. reinforce the motivation for better models of AI's reliability brought to LLMs, and Chen et al.'s paper serves as first steps in the analysis of LLM evolution. These papers are now included in the paper and covered in more detail in appendix A.14 (See also R1.2 here).

⁶⁰ Hoffmann, J., Borgeaud, S., Mensch, A., Buchatskaya, E., Cai, T., Rutherford, E., ... & Sifre, L. (2022). Training compute-optimal large language models. arXiv preprint arXiv:2203.15556.

Reviewer Reports on the First Revision:

Referees' comments:

Referee #2 (Remarks to the Author):

First, I want to heartily applaud the authors on such a substantial revision. It was a joy to read the response letter and see how much work was put in to address reviewer comments.

In particular, the addition of the two human user studies is superb. I will first discuss S2. I would encourage the authors to consider moving the results and some analysis of S2 to the main text. While there has been some prior work which finds that even humans with domain expertise can judge the output of a language model as correct when it's incorrect (e.g., <https://arxiv.org/abs/2306.01694> which I'd encourage the authors to cite), I believe the confusions that the author uncover in many participants' interpretation of LLM responses in S2 is important to communicate to broadly to users of these systems writ large.

To make space for S2 in the main text, I would encourage the authors to move Figure 3 to the Appendix. I don't find Figure 3 particularly informative. And more so, I don't think that key finding F2b can be concluded from the data; to me, it doesn't look like there are substantial differences in the kinds of avoidant behavior as a function of scaling up/shaping up? I would suggest dropping F2b entirely. F2a captures, in my opinion, the key results on avoidance dissipation (which are fascinating, to the authors' credit).

However, I would encourage the authors to modify F2a from "with incorrectness scaling up proportionally" to "ultracrepidarianism scaling up proportionally". My interpretation of Figure 5 is that incorrectness rates for the shaped up models is actually quite stable, but that the rate of avoidance trends down? If that is the correct interpretation, then incorrectness is not actually scaling up? If it's not, the authors may need to do a bit more hand-holding for the reader to clarify.

Now – to S1 (the user difficulty study). I applaud the authors here on undertaking such a substantial overhaul of their original difficulty measure analysis. The new study abates many of my concerns about difficulty metric selection. As a result, I find Figure 2 immensely compelling. To me, Figure 2 is really the banner figure of this paper and will likely draw (worthy) attention from the AI community and beyond.

At the same time, I do have some slight concerns about the particulars of the difficulty measures, some of which may be due to lack of understanding on my part of what was done. If so, I again encourage the authors to clarify their methods. My concerns break down along the following: 1) binning, 2) calibration, and 3) unioning Q1 and Q4. First, my understanding is that the binning (e.g., in the x axis of Fig 2) is intended to ensure equal proportions in each difficulty bucket. However, inspecting the actual bin delimiters (which I found easiest to see in Figure A.19), it looks as though some of the bin endpoints are extremely small, e.g., one angram bin goes from (99.85, 99.99] and another (99.99, 100]. Are the authors sure that these differences are meaningful? For instance, there's a big jump in the angram case between those bins (in Fig A.19); I find this surprising. My

interpretation is just that there happen to be many questions at a high level of difficulty. Would the authors consider coarser bins or perhaps just having all axes on a linear scale from 0 to 100 (given it seems that the authors calibrate difficulty measures to a scale of 0 to 100?)

My confusion here may stem from confusion in the calibration method used. I don't think the authors need to show both raw and calibrated difficulty measures in Table 2. Unless there is something important that the raw scores reveal that the calibrated scores do not, in which case, I think that should be more clearly communicated in the text. It would be nice if the authors include a scatterplot in the Appendix, per task, showing the raw vs. calibrated difficulty scores.

Lastly, I have some concern about the unioning of Q1 and Q4. The authors note in line 942 that they union Q1 and Q4 "to have more responses per item." However, the responses are from the same person! I don't think it is fair then, or gains any meaningful statistical power on sample size, to incorporate two responses from the same person for such similar questions. I would encourage the authors to either just use Q1 or Q4 for their study, or perhaps best, an average of Q1 and Q4 per person (not taking them separately). Nicely, the authors already found that Q1 and Q4 are highly correlated, so I don't think this change should not impact the results, but I believe it is better practice.

Additionally, I have some lingering misgivings in the interpretation of the difficulty scores as a function of the other metrics the authors consider in the paper. I'm surprised that the key findings changed so little with the inclusion of the new results. In particular, my reading of Figure 4 is not necessarily that the medium difficulty scores, when looking at correctness, have the highest levels of prompt sensitivity (implied from the paragraph starting on line 215 and in Key Finding F1b). There are a few instances of this, but generally, my interpretation of the prompt sensitivity plot is that the different models can have markedly different levels of prompt sensitivity, and that this varies across difficulty. This is important for users as, without predictability in whether a model is going to be sensitive to prompts, it may be hard to trust its output. I think the authors could reasonably discuss reliability there, and naturally connect to the results of S2.

Beyond the new user studies, I applaud the authors on the large effort to include two new tasks. I think the incorporation of the "science" and "transform" tasks, particularly the latter, help with generality. My understanding is that the "transforms" task is a new dataset that the authors contribute? It has bits from other datasets, but I would encourage the authors to consider more clearly pitching "transforms" new benchmark, particularly now that the authors have human data for it! As for "science" – I have some concern about possible data contamination here, as the authors note they do too in line 151. If possible, I would encourage the authors to try to conduct some analysis to get a sense for how contaminated the latest models may be on these tasks (this may not be possible, but if it is, could add substantial value). In particular, if the authors do find that there is contamination, that's quite an undercut to the massive increase in correctness that some of the latest models show on science. I think that subfinding may be of nice value as well.

Overall though, I again applaud the authors on a substantial revision. As the authors note in their response letter, the "fast-changing" nature of LLMs demands more comprehensive evaluation studies, of the kinds these authors attempt to address. I do feel there are a few clarity points, as

noted above, that warrant addressing. However, the results - to my understanding - are interesting and important contributions to the broader study of LLM evolution and their use.

Some additional bespoke comments:

- I would reword the difficulty measure not in the limitations [line 294] to mention more that the humans you've recruited are not experts (as part of generality). I don't think that's a huge concern here, but worth caveating that "human difficulty" really depends on which humans you ask!

- I agree with R1's first review that Figure 1 is a bit confusing. The new wording helps, but I actually found Table A.9 a clearer depiction of the trends intended to be show in Figure 1.

- Please include some description of the generality measure for the difficulty concordance score. I was not able to clearly follow what "difficulty concordance" really means here.

- The phrasing in the box starting on line 105 was confusing, especially (1) and (2). Please consider rephrasing (e.g., "so" to "thereby" in (1)? And clarifying the "or" in (2)).

Define Q1-Q4 before Section A.6 (they're only defined later in A.10)

- Minor note for future human studies — S2 could have likely been improved by having participants respond on a slider or Likert scale ranging from "definitely incorrect" to "definitely correct". This would have yielded a richer response. The current abstractly-labeled A, B, C... buttons may have induced more cognitive load on the participant.

“Larger and more instructable language models turned less reliable”

(original title: Scaled-up, Shaped-up, but Letting Down? Reliability Fluctuations of Large Language Model Families)

Revision Letter - June 2024

We thank the editor and reviewer for their feedback in this new iteration. We include a summarised list of changes explaining how we dealt with the editor’s indications, followed by a point-by-point response to the referee’s comments. The letter is closed by details about how we have made our manuscript comply with the formatting requirements.

The authors.

Colour legend for manuscript: *blue (inserted)*, *green (moved)*, *red (modified)*. Deletions simply removed.

Editorial Suggestions

Your manuscript, "Scaled-up, Shaped-up, but Letting Down? Reliability Fluctuations of Large Language Model Families", has now been seen by 1 referee, whose comments are attached below (referee #1 was unavailable this round but we felt this report was sufficient to proceed). While they continue to find your work of interest - and applaud the hard work put into this revision (as indeed do we!), they have raised some final points that need to be addressed before we can make a decision on publication. We will need to consider your response to these concerns in the form of a revised manuscript accompanied by a list to explain your revisions. You will also need to make some editorial changes to your paper so that it is as brief as possible and complies with our Guide to Authors (<https://www.nature.com/nature/for-authors>).

E.1. In particular, they have some clarifications and concerns about the choices made for binning and calibration of some of the metrics reported. And they encourage you to take care in ensuring that the conclusions drawn in the text reflect the newly revised experimental results.

Answer. This is the list of changes, starting with the modifications about binning and calibration.

- We have now explained more clearly how we do calibration. We use the common Platt scaling for mapping values to percentages in calibrated scales that reflect the probability that a human fails. The transformation is a logistic function, so it does not affect the same-width bins (it only affects the numbers on the x-axis, but not the “shape” of all figures and results). This binning shows that for some datasets (notably for addition) we have a high concentration of difficult examples, with half of them of a calibrated difficulty greater than 98.7%. This means that for half of them 98.7% or more of humans are expected to fail according to the perception given by the human sample in our study S1. We have included the calibration functions (Table A.10) and a scatter plot showing the mapping (Figure A.14) in a major extension of appendix A.13 (old appendix A.16). This fully follows the reviewer’s request, and explains binning and calibration much better.
- We have changed the way we obtain the human difficulty reference from S1, using Q1 and Q4, following the reviewer’s suggestion. Now it’s the median of a mean rather than a mean of a median (with swapped groupings). As expected, all correlations and the calibration

function are very similar, but for addition, the “carry” function shows slightly higher correlation than the other difficulty metric solely based on #digits. Both were very close with the previous calculation already, because the number of carry operations is highly correlated with the size of the numbers. So the curves for addition have been replaced with `f_carry` as the difficulty function, even if the changes are almost unnoticeable. Also, the x-axis of several figures have changed slightly because small variations in the calibration functions that use the Q1|Q4 results. This affects Figures 1, 2, 3 (old Figure A.16), Extended Data Figures 3 and 4 (old Figure 4) and several other figures in the appendix, which have been updated.

- We have replaced F2b by a new finding that emphasises the results of S2, as suggested by the reviewer. The title of the manuscript has been replaced by the editor’s suggestion with some minor variations. It is now more factual, integrating all findings opposing the “larger” and more “instructable” model evolution with a reduction of reliability. Figure 4 (old Figure 5) is now using FLOPS instead of only #parameters, and compute has also been added to Table 2, being more comprehensive about the scaling-up part of the paper. The abstract, conclusions and findings have been streamlined and give proportionally more relevance to the restated findings and the human studies.
- We have reduced the size of the paper to around 4000 words and only 6 flying items (figures and tables), streamlining parts of the introduction, results and discussion, with some reorganisation of the material. The image and surrounding text about different kinds of avoidance, as well as old Table 6, have been moved to the appendix (Figure A.19 and Table A.14 respectively) and the prompt sensitivity figures and text moved to extended data (extended data Figures 3 and 4), and also summarised. As suggested, this has made space for some material from the human studies (old Figure A.19 moved to the main paper as Figure 3, alongside a summary of S2 results). Overall, the restructuring ensures we follow the Nature guidelines in terms of structure, size, number of references, etc. We have made some decisions such as keeping Figure 1 in the paper with old Table A.9 moved as Extended Data Table 1, assuming that an online version of the paper would link the two, but we could swap them (e.g., table in the paper and figure in the appendix) if the editor so decides. Figure 2 has fewer panels now (the most representative ones), as suggested, but we have kept the complete figure as an Extended Data Figure 1.
- We have prepared code and data, author contributions and acknowledgements. Some are still not included or fully linked in this revision because of anonymity.

In the end, the paper has now four parts: main, methods, extended data and supplementary material, all of them following the guidelines, but most or even all of the supplementary material could be moved at the end of the methods section if so required.

Referee #2

First, I want to heartily applaud the authors on such a substantial revision. It was a joy to read the response letter and see how much work was put in to address reviewer comments.

R1. In particular, the addition of the two human user studies is superb. I will first discuss S2. I would encourage the authors to consider moving the results and some analysis of S2 to the main text. While there has been some prior work which finds that even humans with domain expertise can judge the output of a language model as correct when it's incorrect (e.g., <https://arxiv.org/abs/2306.01694> which I'd encourage the authors to cite), I believe the confusions that the author uncover in many participants' interpretation of LLM responses in S2 is important to communicate to broadly to users of these systems writ large.

We have moved Figure A.13 in the previous version and its associated analysis to the main text in this version (as Figure 3). This summarises the key results of S2, used to support finding F2b, which replaces the old one (see R2 below). This states that we cannot rely on humans for oversight and supervision. In other words, human intuition on difficulty does not reduce the supervision errors from incorrect to correct. The suggested reference is now also discussed in the section "*Appendix A.13. Extended coverage of state of the art*". Collins et al. found that even humans with domain expertise could judge the output of a language model as correct when it was incorrect, specifically in the context of undergraduate-level theorem proving. Our work is consistent with this, and extends it across a variety of domains. It also differs in that we allow 4-valued selections (correct, incorrect, avoidance and unsure) and make use of human difficulty (a novelty in our work), highlighting a broader issue of reliability with a novel perspective.

R2. To make space for S2 in the main text, I would encourage the authors to move Figure 3 [now Figure A.19] to the Appendix. I don't find Figure 3 particularly informative. And more so, I don't think that key finding F2b can be concluded from the data; to me, it doesn't look like there are substantial differences in the kinds of avoidant behavior as a function of scaling up/shaping up? I would suggest dropping F2b entirely. F2a captures, in my opinion, the key results on avoidance dissipation (which are fascinating, to the authors' credit).

However, I would encourage the authors to modify F2a from "with incorrectness scaling up proportionally" to "ultracrepidarianism scaling up proportionally". My interpretation of Figure 5 [Now Figure 4] is that incorrectness rates for the shaped up models is actually quite stable, but that the rate of avoidance trends down? If that is the correct interpretation, then incorrectness is not actually scaling up? If it's not, the authors may need to do a bit more hand-holding for the reader to clarify.

Answer. As suggested, we have moved Figure 3 (old numbering) and the accompanying discussion to the Appendix as Figure A.19 (jointly with the avoidance type rubric already in the appendix), and referred to it in the main text. We have replaced F2b by a new finding emphasising the results of S2 (see response R1 above), highlighting the lack of concordance between avoidance and difficulty, an important finding that remains relevant as human users would expect

such a relation to exist, but surprisingly this expectation is not used by humans to reduce the incorrect-to-correct kind of supervision errors.

Regarding Figure 5 (now Figure 4) and the incorrectness rates for shaped-up models, we thank the referee for catching this unclarity. We meant 'proportionally to the number of not-correct answers' (i.e., ultracrepidarianism) and have adapted the respective statement as "scaling and shaping currently trades avoidance for more incorrectness" (F2a).

R3. Now – to S1 (the user difficulty study). I applaud the authors here on undertaking such a substantial overhaul of their original difficulty measure analysis. The new study abates many of my concerns about difficulty metric selection. As a result, I find Figure 2 immensely compelling. To me, Figure 2 is really the banner figure of this paper and will likely draw (worthy) attention from the AI community and beyond.

Answer. Thank you.

R4. At the same time, I do have some slight concerns about the particulars of the difficulty measures, some of which may be due to lack of understanding on my part of what was done. If so, I again encourage the authors to clarify their methods. My concerns break down along the following: 1) binning, 2) calibration, and 3) unioning Q1 and Q4. First, my understanding is that the binning (e.g., in the x axis of Fig 2) is intended to ensure equal proportions in each difficulty bucket. However, inspecting the actual bin delimiters (which I found easiest to see in Figure A.19 [Now Figure 3]), it looks as though some of the bin endpoints are extremely small, e.g., one anagram bin goes from (99.85, 99.99] and another (99.99, 100]. Are the authors sure that these differences are meaningful? For instance, there's a big jump in the anagram case between those bins (in Fig A.19); I find this surprising. My interpretation is just that there happen to be many questions at a high level of difficulty. Would the authors consider coarser bins or perhaps just having all axes on a linear scale from 0 to 100 (given it seems that the authors calibrate difficulty measures to a scale of 0 to 100?)

My confusion here may stem from confusion in the calibration method used. I don't think the authors need to show both raw and calibrated difficulty measures in Table 2. Unless there is something important that the raw scores reveal that the calibrated scores do not, in which case, I think that should be more clearly communicated in the text. It would be nice if the authors include a scatterplot in the Appendix, per task, showing the raw vs. calibrated difficulty scores.

Answer. Calibration is performed with 90 instances per benchmark originating from survey S1. Prior to binning, we fit a logistic function with two parameters. This is actually a very common calibration method known as Platt scaling, but we didn't properly acknowledge this in the previous version. Now we do, and explain more clearly in the methods section that there is this parametric function that maps the raw difficulty that comes from the benchmark to a scale between 0 and 100 that corresponds to the probability of human failure as estimated by humans themselves. We also state very clearly now that because of the use of a human sample and the parametric fit, small differences in the scale should be interpreted with care.

In appendix A.8, we now give the slope and position for each of these calibration functions (Table A.10) and, as suggested, we use scatter plots (now included in Fig. A.14 in appendix A.8) with the original difficulties of all instances mapped to the calibrated ones using this logistic function. This shows the parts of the x-axis and y-axis where points are most compacted, especially those around 100% on the y-axis (for addition, not anagram), because of the effect of the logistic function

and also the distribution of difficulties. These narrower bins illustrate what the reviewer says more clearly now: for some datasets there's a high concentration of examples for which most humans are expected to fail (large additions, locations in general, science questions for graduate students, etc.). We include the 30 bins and 5 bins arrangements in these plots, showing the mean and the standard deviation in each bin, also to see the sensitivity. In some cases the points are very close and the bin limits have to be taken as merely indicative.

About these choices of bins, in Figure 2 and many other figures of the paper, we used 30 bins as this gives smooth "curves" because we have all the examples from the datasets. For Figure 3 (previously numbered as Figure A.19), however, we chose 5 bins because it's a small selection of examples coming from the questionnaire in S2. Using a uniform scale from 0 to 100 would be more intuitive if we could have sampled with uniform difficulty in the first place, but given the datasets and the calibrated difficulties, we have ranges for which we do not have many examples. This also supports the analysis with same-size bins. The use of the calibrated metric in the x-axis just tries to signal all this; there are more examples of some ranges of human-calibrated difficulty than others, but these values are approximate. We have made this more clear in the text.

We also now emphasise in the paper that the calibration does not change the binnings since these are same-size (same proportion as the reviewer says), so the bin limits that we show between 0% and 100% are an estimate of human success (or rather expected failure %), and the accuracy of these percentages may be low (given the sample and how narrow some bins are). But even if the bin limits were not reliable, the bins would be the same for the two scales (raw and calibrated), as the calibration is monotonic.

Since the appendix now gives a complete account of the calibration, we have now removed the raw difficulties from Table 2.

R5. Lastly, I have some concern about the unioning of Q1 and Q4. The authors note in line 942 that they union Q1 and Q4 "to have more responses per item." However, the responses are from the same person! I don't think it is fair then, or gains any meaningful statistical power on sample size, to incorporate two responses from the same person for such similar questions. I would encourage the authors to either just use Q1 or Q4 for their study, or perhaps best, an average of Q1 and Q4 per person (not taking them separately). Nicely, the authors already found that Q1 and Q4 are highly correlated, so I don't think this change should not impact the results, but I believe it is better practice.

Answer. Thank you for this indication. For $Q1 \cup Q4$, we actually calculated the median for all individuals per instance and question. And we built the mean between Q1 and Q4 for each instance a posteriori. But it makes more sense, as the reviewer says, to calculate the mean between Q1 and Q4 for each individual and then calculate the median of this new "Q1|Q4" for all individuals doing the item. We have explained that this choice is made for robustness per individual and not "to have more responses per item".

With the new calculation of this Q1|Q4 we have very similar results, as anticipated (it's now a median of a mean rather than a mean of a median with a different grouping). Correlations are very similar, but the two most correlated difficulty functions for addition are now swapped: the carry difficulty function is now slightly more correlated than the $\min(\#\text{digits})$. Apart from this change of the selected difficulty function for addition, which is now reflected in the paper, all other changes are very small or unnoticeable, but still we have modified all procedures and plots that depend on this (calibration, figures, tables, etc.).

R6. Additionally, I have some lingering misgivings in the interpretation of the difficulty scores as a function of the other metrics the authors consider in the paper. I'm surprised that the key findings changed so little with the inclusion of the new results. In particular, my reading of Figure 4 [Now Extended Data Figures 3 and 4] is not necessarily that the medium difficulty scores, when looking at correctness, have the highest levels of prompt sensitivity (implied from the paragraph starting on line 215 and in Key Finding F1b). There are a few instances of this, but generally, my interpretation of the prompt sensitivity plot is that the different models can have markedly different levels of prompt sensitivity, and that this varies across difficulty. This is important for users as, without predictability in whether a model is going to be sensitive to prompts, it may be hard to trust its output. I think the authors could reasonably discuss reliability there, and naturally connect to the results of S2.

Answer. Accordingly, we have made changes in both the last paragraph of page 6 (line 215 in the previous submission) and the key finding F3a (which the reviewer seems to have mistyped as F1b), and naturally connecting all this to reliability and the results of S2.

R7. Beyond the new user studies, I applaud the authors on the large effort to include two new tasks. I think the incorporation of the “science” and “transform” tasks, particularly the latter, help with generality. My understanding is that the “transforms” task is a new dataset that the authors contribute? It has bits from other datasets, but I would encourage the authors to consider more clearly pitching “transforms” new benchmarks, particularly now that the authors have human data for it! As for “science” – I have some concern about possible data contamination here, as the authors note they do too in line 151. If possible, I would encourage the authors to try to conduct some analysis to get a sense for how contaminated the latest models may be on these tasks (this may not be possible, but if it is, could add substantial value). In particular, if the authors do find that there is contamination, that's quite an undercut to the massive increase in correctness that some of the latest models show on science. I think that subfinding may be of nice value as well.

Answer. Thank you. We have now pitched “transform” as a new benchmark, highlighting that it has human data for calibrating difficulties and supervision (although all datasets and human data will also be available with the paper), in the Data Generation part of the Methods section. Regarding contamination, we now explicitly discuss this in a new appendix A.14 with new Table A.15. There we show that contamination is unlikely for ‘science’ and at most partial for ‘transforms’, but is very unlikely for the other datasets, as they're newly introduced in this work. For ‘science’: (i) the GPQA part of ‘science’ was released after all the studied LLMs were released, and thus unlikely to have contamination; (ii) for OpenBookQA part of ‘science’, we found a recent work investigating contamination (<https://aclanthology.org/2024.naacl-long.482/>), which has reported relatively low risk of contamination on OpenBookQA in comparison with many other popular benchmarks. Thus, the risk of contamination for ‘science’ seems low, though this does not completely rule out the possibility. We still think the massive increase in correctness for some of the latest LLMs may have more to do with intrinsic factors in alignment with the other datasets. For ‘transforms’, there was a subset of ‘data wrangling’ tasks that were included in BigBench, released in June 2022 (and parts of it even earlier). Unlike OpenBookQA, no prior work has analysed data contamination of the ‘datawrangling’ tasks, likely because the outputs are open-ended, for which detecting data contamination is more challenging than multiple-choice questions. Thus, we cannot rule out the possibility that there may be contamination. The new Table A.12 summarises our estimates of possible contamination. Overall, only a sample of one single benchmark (among the five that we analyse) may be contaminated.

R8. Overall though, I again applaud the authors on a substantial revision. As the authors note in their response letter, the “fast-changing” nature of LLMs demands more comprehensive evaluation studies, of the kinds these authors attempt to address. I do feel there are a few clarity points, as noted above, that warrant addressing. However, the results - to my understanding - are interesting and important contributions to the broader study of LLM evolution and their use.

Answer. We thank the reviewer for these kind words, and we likewise extend our appreciation for the in-depth and on-point comments made by the reviewer, which raised the quality of this paper across the board.

R9. Some additional bespoke comments:

- I would reword the difficulty measure not in the limitations [line 294] to mention more that the humans you’ve recruited are not experts (as part of generality). I don’t think that’s a huge concern here, but worth caveating that “human difficulty” really depends on which humans you ask!

Answer. Accordingly, we have reworded the sentences (line 294 from the previous submission, now in the last paragraph of the paper) to note that human difficulty depends on the sample of humans that are asked.

R10. - I agree with R1’s first review that Figure 1 is a bit confusing. The new wording helps, but I actually found Table A.9 a clearer depiction of the trends intended to be show in Figure 1.

Answer. Debating between swapping Figure 1 and Table A.9 (now as Extended Data Table 1), we prefer the visual representation of Figure 1 as the entry point to the summarised results of the paper. We now explain in Figure 1 that the table gives a more detailed and regular perspective.

R11. - Please include some description of the generality measure for the difficulty concordance score. I was not able to clearly follow what “difficulty concordance” really means here.

Answer. The description of the generality metric has been expanded in the Methods section.

R12. - The phrasing in the box starting on line 105 was confusing, especially (1) and (2). Please consider rephrasing (e.g., “so” to “thereby” in (1)? And clarifying the “or” in (2)). Define Q1-Q4 before Section A.6 (they’re only defined later in A.10)

Answer. We have rephrased the corresponding text and moved the sections dealing with the human study around to be in order and closer together. They are now appendices A.5 to A.7 and ensure Q1-Q4 are defined before.

R13. - Minor note for future human studies — S2 could have likely been improved by having participants respond on a slider or Likert scale ranging from “definitely incorrect” to “definitely correct”. This would have yielded a richer response. The current abstractly-labeled A, B, C... buttons may have induced more cognitive load on the participant.

Answer. We would like to note that a slider or Likert scale by itself would not fit a 3-valued classification (incorrect, avoidant, correct), as avoidance does not fit on the incorrect-correctness spectrum. We do agree that we should have labelled the abstract ABC options with their respective value, and thank the reviewer for this suggestion on future work.

Reviewer Reports on the Second Revision:

Referees' comments:

Referee #2 (Remarks to the Author):

I again applaud the authors on excellent revisions and a thorough point-by-point response. I believe the changes have substantially improved the paper, and I have no major outstanding concerns at this time.